# SATURN: SAT-based Reinforcement Learning to Unleash Language Model Reasoning

**Huanyu Liu**
Peking University
huanyuliu@stu.pku.edu.cn

**Ge Li**[*]
Peking University
lige@pku.edu.cn

**Jia Li**
Tsinghua University
jia_li@mail
.tsinghua.edu.cn

**Hao Zhu**
Peking University
zhuhao@stu.pku.edu.cn

**Kechi Zhang**
Peking University
zhangkechi@pku.edu.cn

**Yihong Dong**
Peking University
dongyh@stu.pku.edu.cn

## Abstract

How to design reinforcement learning (RL) tasks that effectively unleash the reasoning capability of large language models (LLMs) remains an open question. Existing RL tasks (*e.g.,* math, programming, and constructing reasoning tasks) face three key limitations: ❶ **Scalability**. They rely heavily on human annotation or expensive LLM synthesis to generate sufficient training data. ❷ **Verifiability**. LLMs' outputs are hard to verify automatically and reliably. ❸ **Controllable Difficulty**. Most tasks lack fine-grained difficulty control, making it challenging to train LLMs from easy to hard and progressively develop reasoning capability.

To address these limitations, we propose **SATURN**, a SAT-based RL framework that uses Boolean Satisfiability (SAT) problems to train and evaluate LLM reasoning. **SATURN** enables scalable task construction, rule-based verification, and precise difficulty control. **SATURN** designs a curriculum learning pipeline that continuously improves LLMs' reasoning capability by constructing SAT tasks of increasing difficulty and training LLMs from easy to hard. To ensure stable training, we design a principled mechanism to control difficulty transitions.

We introduce **SATURN-2.6k**, a dataset of 2,660 SAT problems with varying difficulty. It supports the evaluation of how LLM reasoning changes with problem difficulty. We apply **SATURN** to DeepSeek-R1-Distill-Qwen and obtain **SATURN**-1.5B and **SATURN**-7B. We achieve several notable results: ❶ On SAT problems, **SATURN**-1.5B and **SATURN**-7B achieve average `pass@3` improvements of +14.0 and +28.1, respectively. ❷ On math and programming tasks, **SATURN**-1.5B and **SATURN**-7B improve average scores by +4.9 and +1.8 on benchmarks (*e.g.,* AIME, LiveCodeBench). ❸ Compared to the state-of-the-art (SOTA) approach in constructing RL tasks, **SATURN** achieves further improvements of +8.8%. We release the source code, data, and models to support future research at `https://github.com/gtxygyzb/Saturn-code`.

## 1 Introduction

Recently, reinforcement learning (RL) has become a promising paradigm for unleashing the reasoning capability of large language models (LLMs), particularly in math, programming, and logical reasoning (*e.g.,* OpenAI-o1 [21], DeepSeek-R1 [11], Kimi-k1.5 [37]). During the RL training process, the

---

[*]Corresponding author.

39th Conference on Neural Information Processing Systems (NeurIPS 2025).

design of RL tasks plays a critical role [14, 31, 32]. A well-designed RL task should elicit LLMs' reasoning capability, fostering behaviors such as hesitation, reflection, backtracking, summarization, and verification [28, 31, 43, 44, 53].

However, how to design RL tasks that can continuously enhance LLMs' reasoning capability remains an open question. We think a well-designed RL task for reasoning should satisfy the following three criteria: ❶ **Scalability.** RL training requires large-scale data. RL tasks should support scalable data without human annotation or expensive LLMs' synthesis. ❷ **Verifiability.** RL rewards must be unambiguously correct. The outputs of LLMs for the task should be easy to verify. ❸ **Controllable Difficulty.** Reasoning capability emerges progressively [42]. RL tasks should support the difficulty control to enable curriculum learning, allowing LLMs to gradually develop more complex reasoning skills [16].

Table 1: The comparison between existing RL tasks and **SATURN**.

| Tasks | Scalability | Verifiability | Controllable Difficulty |
|---|---|---|---|
| ScaleQuest [12] | ✗ | ✗ | ✗ |
| GSM8K (Math) [9] | ✗ | ✓ | ✗ |
| LiveCodeBench [22] | ✗ | ✓ | ✗ |
| Game Werewolf [45, 48] | ✗ | ✗ | ✗ |
| LMRL Gym [4] | ✗ | ✓ | ✓ |
| SPAG [7] | ✗ | ✓ | ✗ |
| Knights&Knaves [46] | ✓ | ✓ | ✗ |
| **SATURN** (Ours) | ✓ | ✓ | ✓ |

Table 1 shows the features of current mainstream RL tasks. None of them satisfy all three criteria. Existing RL tasks can be divided into two categories: (1) One category of RL tasks requires LLMs to solve math or programming problems, with rewards based on the correctness of the final answer or code [5, 6, 9, 27]. However, these tasks rely on human annotation for ground-truth solutions or test cases, suffer from a lack of high-quality problems, and offer only coarse control over reasoning difficulty [23, 26]. (2) Another category focuses on manually designed reasoning tasks [12, 45, 46, 48]. For instance, Logic-RL [47] leverages natural language logic K&K puzzles to improve LLMs' reasoning capability through RL. However, they also present limitations, such as hard to scale up due to reliance on sampling from LLMs [7, 12], hard to verify despite relying on LLMs for cross-validation [12, 41, 51], and hard to control difficulty [45, 48].

In this paper, we aim to answer the following research question:

> **Key Question**
>
> Can we design an RL task that supports scalability, verifiability, controllable difficulty, and enhances the reasoning capability of LLMs?

To this end, we propose **Boolean Satisfiability (SAT) problem** as the task for RL. Figure 1 shows an illustration of SAT problems and corresponding features. SAT satisfies all three desiderata we outlined earlier: ❶Scalability. SAT instances can be generated programmatically at scale without human annotation or LLM synthesis, allowing for virtually unlimited training data. ❷ Verifiability. SAT is a well-established NP-complete problem in theoretical computer science [10]. The correctness of a solution can be easily verified in linear time. But solving SAT problems requires complex reasoning. ❸ Controllable Difficulty. The difficulty of SAT instances can be precisely adjusted (*e.g.,* number of variables, clauses), making it suitable for curriculum learning. What's more, SAT serves as a universal substrate for limited forms of logical reasoning, as many problems in propositional logic, finite-domain first-order logic, and modal logic can be systematically reduced to SAT [17, 24, 36].

Building on these advantages, we propose _**SAT**-based reinforcement learning to **U**nleash LLMs **R**easo**N**ing_, or **SATURN**. **SATURN** is a multi-stage curriculum learning-based RL framework that continuously improves the reasoning capability of LLMs. **SATURN** efficiently constructs SAT tasks with controllable difficulty and organizes them into progressive stages from easy to hard, allowing LLMs to develop reasoning skills step by step. To ensure training stability and effective progression, we design a principled mechanism to control difficulty transitions based on LLMs' performance. **SATURN** enables smooth curriculum advancement and steady enhancement of reasoning capability.

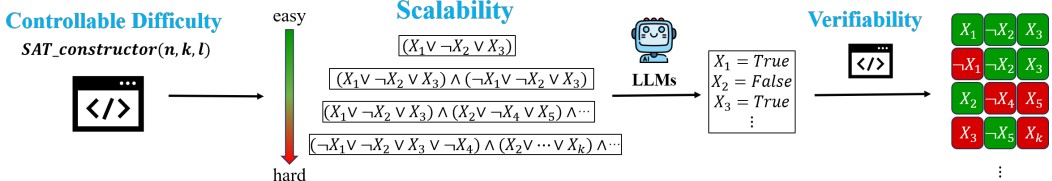

Figure 1: An illustration of SAT problems and its corresponding features.

We introduce the **SATURN-2.6k** dataset, consisting of 1,500 training instances, 160 test instances at the same difficulty as the training set, and 1,000 test instances from 10 **harder** unseen difficulty levels. The test set serves as a benchmark for systematically evaluating how LLMs' reasoning capability varies with increasing SAT task difficulty. We release SAT construction scripts alongside the dataset, which enable the creation of virtually unlimited SAT instances.

We apply **SATURN** to DeepSeek-R1-Distill-Qwen-1.5B and 7B [11], obtaining **SATURN**-1.5B and **SATURN**-7B. Experiments show that **SATURN** effectively enhances LLMs' reasoning capability in generalizable scenarios:

- **SATURN**-1.5B and **SATURN**-7B achieve strong performance on **SATURN-2.6k** benchmarks. On unseen harder test set, two models achieve `pass@3` improvements of +14.0 and +28.1 respectively.
- The reasoning capability learned from **SATURN** transfers well to math and programming tasks, bringing average improvements of +4.9 and +1.8 on popular benchmarks such as AIME [2], AMC [1], MATH-500 [19], GPQA Diamond [35], and LiveCodeBench [22] for two **SATURN** models.
- Compared to the prior SOTA approach (*e.g.,* Logic-RL), **SATURN** achieve average improvements of +8.8% on math and programming tasks.

## 2 SATURN

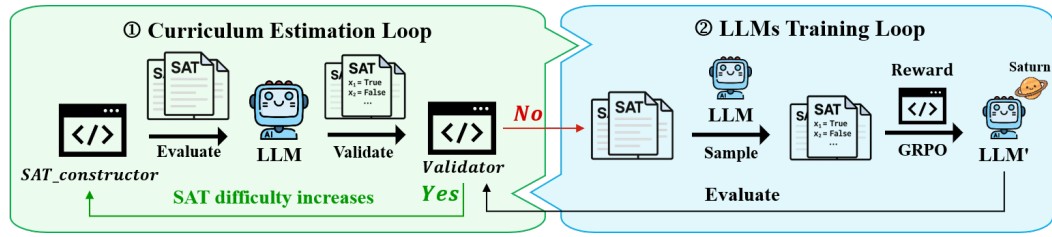

Figure 2: The overall framework of **SATURN**. It alternates between two interconnected loops: (1) **Curriculum Estimation Loop.** (2) **LLMs Training Loop.** The two loops iterate until the maximum number of curriculum stages is reached.

### 2.1 SATURN Learning Loop Framework

We introduce **SATURN**, a multi-stage RL framework that leverages SAT tasks to unleash LLMs' reasoning via curriculum learning. As illustrated in Figure 2, **SATURN** alternates between two interconnected loops: *Curriculum Estimation Loop* dynamically constructs SAT instances of adjustable difficulty and evaluates LLMs' performance to determine whether to advance the curriculum stage; *LLMs Training Loop* employs RL to optimize LLMs on current difficulty SAT tasks. The curriculum loop presented in Algorithm 1 proceeds as follows:

**Step 1: Curriculum Estimation Loop.** Given initial SAT difficulty, `SAT_Construction` generates a validation set of SAT instances. The LLM is evaluated on this set using the `pass@1` metric. If the performance exceeds a predefined threshold $\epsilon$, the curriculum controller advances to a harder configuration with an increased estimated difficulty. Otherwise, **SATURN** process enters **Step 2** LLMs training loop at the current SAT difficulty. This adaptive loop ensures that the LLM is always trained at the frontier of its reasoning capability, neither too easy nor too hard.

**Step 2: LLMs Training Loop.** For the current difficulty, `SAT_Construction` generates a set of training instances that are different from the validation set. These samples are then used to train LLMs with GRPO. The reward function encourages outputs that are both logically correct and properly formatted. The training loop proceeds until `pass@1 > ε` on the validation set. After that, the process backs to **Step 1** to reassess and potentially advance the curriculum.

The two loops iterate jointly. **SATURN** process terminates when a predefined total number of iterations is reached. Importantly, **SATURN** is not designed to replace math or programming tasks, but to serve as a complementary strategy for enhancing LLMs' reasoning. In practice, **SATURN** can be integrated with math and programming tasks to enable a stronger training framework.

**SATURN** learning loop raises three core challenges: ❶ Section 2.2 introduces how to construct scalable and controllable SAT instances. ❷ Section 2.3 presents how to estimate instance difficulty for curriculum learning. ❸ Section 2.4 explains how to train LLMs on SAT tasks with RL.

## 2.2 SAT Instances Construction

In this subsection, we formalize the construction of SAT instances. A SAT problem determines whether a propositional formula can be satisfied by a Boolean truth assignment. Formally, we define a $(n, k, l)$-SAT instance in conjunctive normal form (CNF) as:

$$
\begin{cases}
\Phi = \left(x_{a_{1,1}} \vee \neg x_{a_{1,2}} \vee \cdots \vee x_{a_{1,n}}\right) \wedge \cdots \wedge \left(x_{a_{l,1}} \vee \cdots \vee \neg x_{a_{l,n}}\right) \\
\text{where } a_{i,j} \in \{1, \ldots, k\}, \quad i \in [1, l]_{\mathbb{Z}}, j \in [1, n]_{\mathbb{Z}}
\end{cases}
\tag{1}
$$

where each clause contains exactly $n$ variables (literals), each being either $x_i$ or its negation $\neg x_i$, $k$ is the total number of variables, and $l$ is the total number of clauses. Based on the definition, we design a SAT instance constructor, `SAT_Construction(n, k, l, m)`, which uniformly samples $m$ SAT instances from the space of $(n, k, l)$-SAT. By adjusting the parameters $(n, k, l, m)$, `SAT_Construction` enables the scalable and controllable construction of SAT instances. The design details of the constructor algorithm are provided in Appendix B. All generated SAT instances are guaranteed to be satisfiable.

## 2.3 Estimation of Task Difficulty

In this subsection, we present the estimation of SAT task difficulty for LLMs. This estimation also serves as the foundation for curriculum learning in LLMs.

As a canonical NP-complete problem [10], SAT admits a polynomial-time reduction from any other NP problem [25]. SAT exhibits a known phase transition phenomenon: when the clause-to-variable ratio $\alpha_c = l/k$ approaches a critical threshold (typically near $4.26$ for 3-SAT), the probability of satisfiability drops sharply, and problem difficulty peaks. This phenomenon probably stems from *replica symmetry breaking* (RSB) [54]: near $\alpha_c$, the solution space fractures into disconnected clusters separated by energy barriers. Beyond $\alpha_c$, the space collapses, reducing complexity.

However, RSB theory is designed for heuristic SAT solvers. For humans or LLMs solving SAT problems through logical steps such as trial, verification, and reasoning, such solver-like phase transitions are hardly observable in human-like thinking processes. While any $n$-SAT ($n > 3$) can be reduced to 3-SAT [25], they differ significantly for LLMs in terms of solution space size and token length.

Prior work [18] on SAT tasks for LLMs typically categorized difficulty based on phase transition points. To systematically estimate task difficulty, we define an analytical estimator of the expected solution space size. Given a $(n, k, l)$ SAT instance, its difficulty for LLMs can be approximately estimated by:

$$
D(n, k, l) = \log_2(k) + 2\log_2(l) - n + \frac{k}{n}
\tag{2}
$$

Eq. (2) provides a more controllable, fine-grained estimation of SAT task difficulty. The detailed derivation is provided in Appendix C. To further validate Eq. (2), we evaluate LLMs' performance on SAT instances with varying difficulty levels. As shown in Figure 3, each point represents a LLM's

average `pass@3` on the same estimated difficulty instances. `Pass@3` generally decreases as $D(n,k,l)$ increases, suggesting that our estimation aligns with the solvability trends observed in practical LLMs. Stronger LLMs maintain higher `pass@3`, while weaker LLMs exhibit lower scores overall. The validity of the estimation in Eq. (2) is further confirmed by ablation experiments, as detailed in Appendix C.

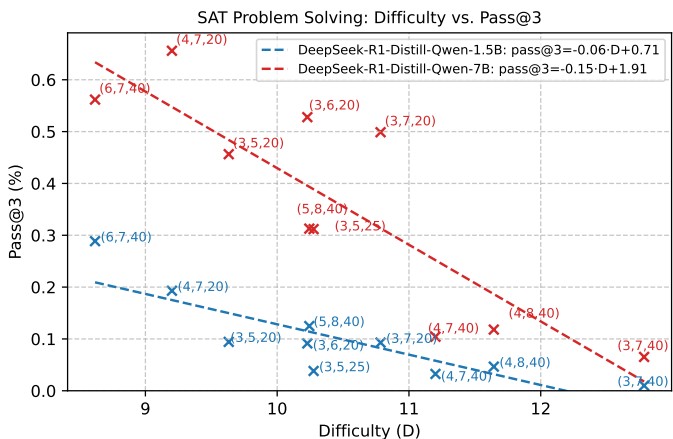

Figure 3: Scatter plots of `pass@3` versus estimated difficulty $D(n,k,l)$ for different LLMs, with linear regression fits. The linear regression for two models achieve $R^2$ values of 0.707 and 0.724 respectively, suggesting a reasonably strong linear relationship between difficulty and `pass@3`.

## 2.4 Reinforcement Learning with GRPO

In this subsection, we introduce the single-stage RL training for given $(n,k,l)$-difficulty tasks. RL can further improve LLMs' generalization by directly optimizing policy gradients over diverse reasoning trajectories [11]. Given the SAT tasks, we then train LLMs using the original sample-level GRPO to optimize the policy $\pi_\theta$ with KL divergence penalty. The GRPO objective function is defined as:

$$
\begin{aligned}
\mathcal{L}_{\text{GRPO}}(\theta) =& \mathbb{E}[q \sim P(Q), \{o_i\}_{i=1}^{G} \sim \pi_{\theta_{old}}(O|q)] \\
& \frac{1}{G}\sum_{i=1}^{G}\frac{1}{|o_i|}\sum_{t=1}^{|o_i|}\Big\{\min\Big[r_{i,t}(\theta)\hat{A}_{i,t}, \text{clip}\left(r_{i,t}(\theta), 1-\epsilon, 1+\epsilon\right)\hat{A}_{i,t}\Big] - \beta\mathbb{D}_{\text{KL}}\left[\pi_\theta \parallel \pi_{\text{ref}}\right]\Big\} \\
\text{where } & r_{i,t}(\theta) = \frac{\pi_\theta(o_{i,t} \mid q, o_{i,<t})}{\pi_{\theta_{old}}(o_{i,t} \mid q, o_{i,<t})}, \quad \hat{A}_{i,t} = \frac{r_i - \text{mean}(\mathbf{r})}{\text{std}(\mathbf{r})}
\end{aligned}
\tag{3}
$$

where $q$ denotes a SAT instance, $o_i$ is the reasoning trajectory generated by the policy $\pi_\theta$, and $G$ groups SAT instances with identical $(n,k,l)$ parameters. A simple yet effective reward scheme [11] is designed that combines a *format reward* and a *correctness* reward. Specifically, $r_i = -1$ if an output is invalid (*i.e.,* missing the \boxed{} wrapper); $r_i = 0$ for well-formatted but incorrect answers; and $r_i = 1$ only when both the format and the answer are correct. Here, an answer is considered correct if it passes a verifier and represents a full satisfying assignment. Training schedule and hyperparameter settings are detailed in Appendix D. And the SAT prompt template is shown in Appendix F.

## 3 Experiments

We apply **SATURN** to DeepSeek-R1-Distill-Qwen-1.5B and 7B, obtaining **SATURN**-1.5B and **SATURN**-7B. To evaluate the effectiveness of **SATURN**, we conduct a large-scale study to evaluate both models. In this section, we introduce our research questions (RQs), benchmarks, baselines, and evaluation metrics. For each RQ, we present the corresponding experimental design, results, and analysis in separate subsections.

## 3.1 Research Questions

Our study aims to answer the following research questions:

**RQ1: How much improvement does SATURN achieve in solving SAT tasks?** We evaluate **SATURN**-1.5B and **SATURN**-7B performance on **SATURN-2.6k** with different difficulty levels.

**RQ2: How effectively does SATURN generalize to math and programming tasks?** To evaluate the transferability of reasoning capabilities learned by **SATURN**, we evaluate the performance of LLMs on math and programming benchmarks and compare them with current SOTA LLMs.

**RQ3: How does SATURN compare to prior RL tasks?** To explore the relationship between **SATURN** and existing RL tasks, RQ3 investigates whether **SATURN** can (1) serve as a complementary task to math and programming, and (2) outperform other constructing RL tasks.

**RQ4: How does SATURN affect LLMs reasoning trajectory?** RQ4 explores whether **SATURN** influences the reasoning patterns of LLMs, particularly in terms of response length and the capability of verification. We investigate whether the reasoning improvements observed in SAT tasks generalize to math and programming.

## 3.2 Experimental Setup

**SATURN Hyperparameters** For **SATURN**-1.5B and **SATURN**-7B, we set the initial SAT instance parameters $(n, k, l)$ to $(3, 5, 5)$ and $(3, 5, 13)$, respectively. In *Curriculum Estimation Loop*, the $\epsilon$ threshold is set to 0.5 for the 1.5B model and 0.75 for the 7B model. In *LLMs Training Loop*, we evaluate the `pass@k` with a step size of 250 training samples. The total number of curriculum iterations is set to 2. Detailed hyperparameters are provided in Appendix A. Ablation studies in Appendix H demonstrate the necessity of curriculum learning and the effectiveness of hyperparameters on SAT difficulty, thresholds, step sizes, etc.

**Benchmarks.** ❶ Building upon `SAT_Construction` tool and difficulty estimation, we release **SATURN-2.6k**, a curated benchmark designed to evaluate LLMs' reasoning capability across varying complexity. **SATURN-2.6k** consists of 1,500 training instances and 160 test instances that share the same estimated difficulty level. To assess performance under increasing task complexity, **SATURN-2.6k** further includes 1,000 test instances from 10 unseen harder difficulty levels, with 100 instances per level. These levels are selected based on our difficulty estimation $D(n, k, l)$, enabling a systematic analysis of how LLM performance changes as problem difficulty increases. Additionally, custom datasets of desired difficulty can be constructed using our open-sourced `SAT_Construction` tool. ❷ For math and programming tasks, following DeepSeek-AI [11], we use **AIME 24/25** [2], **AMC 22/23** [1], **MATH-500** [19], **GPQA Diamond** [35], and **LiveCodeBench** `v4_v5` subset [22].

**Baseline Model.** We conduct evaluations against several 1.5B and 7B parameter reasoning models as the baselines, which include DeepSeek-R1-Distill-Qwen-1.5B & 7B [11], Still-3-1.5B-Preview [40], s1.1-1.5B & 7B [31], z1-7B [50], OpenThinker-7B [38], and DeepScaleR-1.5B-Preview [29]. In addition, we include a supervised fine-tuning (SFT)-only baseline trained on the Math training dataset [19], which provides step-by-step problem reasoning trajectories. We randomly select the most difficult Level-5 1,000 problems from training set for one epoch of SFT, following the same training template as DeepSeek-R1-Distill-Qwen. With the same dataset size, our setup enables a fair comparison between SFT and RL on SAT tasks.

**Evaluation Metrics.** Following DeepSeek-AI [11], we use `pass@k` as the evaluation metric. `Pass@k` assesses the probability that at least one correct solution is generated within $k$ attempts. For SAT problems, we evaluate `pass@k` $\in \{1, 3, 5, 7, 10\}$ and sample 12 times per problem. For math and programming benchmarks, we use `pass@1`, following a context length of 32,768 and temperature = 0.6. More evaluation hyperparameters are provided in Appendix E. All experiments are conducted on NVIDIA 8×A100 (40GB) GPUs. Specific prompts are detailed in Appendix F.

## 3.3 RQ1: SATURN Substantially Improves Performance on SAT Tasks

We evaluate the performance of **SATURN**-1.5B and **SATURN**-7B on SAT tasks using **SATURN-2.6k** test set. Specifically, the evaluation involves unseen SAT instances that were not included in the

training data. The results, presented in Table 2, 10–14, and detailed in Appendix G, demonstrate the performance of LLMs across different SAT difficulties.

Table 2: Performance (**pass@k**, in %) on **SATURN-2.6k** test set across different difficulty levels.

| Model | SAT-(3,5,5) | | | | | SAT-(3,5,8) | | | | |
|---|---|---|---|---|---|---|---|---|---|---|
| | @1 | @3 | @5 | @7 | @10 | @1 | @3 | @5 | @7 | @10 |
| DeepSeek-R1-Distill-Qwen-1.5B | 36.7 | 71.7 | 85.4 | 91.7 | 96.2 | 20.3 | 47.6 | 63.6 | 73.4 | 81.9 |
| **SATURN**-1.5B-Iteration-1 | 59.7 | 90.4 | 97.1 | 99.1 | 99.8 | 41.0 | 74.0 | 85.6 | 91.1 | 95.6 |
| **SATURN**-1.5B-Iteration-2 | **70.3** | **95.9** | **99.0** | **99.7** | **99.9** | **47.0** | **82.6** | **93.9** | **98.0** | **99.8** |

| Model | SAT-(3,5,13) | | | | | SAT-(3,5,15) | | | | |
|---|---|---|---|---|---|---|---|---|---|---|
| | @1 | @3 | @5 | @7 | @10 | @1 | @3 | @5 | @7 | @10 |
| DeepSeek-R1-Distill-Qwen-7B | 53.9 | 86.2 | 94.2 | 97.3 | 99.3 | 39.3 | 74.9 | 88.3 | 94.3 | 98.3 |
| **SATURN**-7B-Iteration-1 | 73.0 | 96.1 | 98.9 | 99.7 | 99.9 | 65.7 | 91.8 | 96.8 | 98.7 | 99.7 |
| **SATURN**-7B-Iteration-2 | **89.5** | **99.0** | **99.9** | **100.0** | **100.0** | **85.4** | **98.3** | **99.8** | **99.9** | **100.0** |

**SATURN substantially improves LLM performance on SAT tasks across varying difficulty levels.** On the difficulty SAT-(3,5,5), **SATURN**-1.5B improves pass@1 from 36.7 to 59.7 at Iteration-1, and further to 70.3 at Iteration-2, achieving a total gain of +33.6. On the unseen harder test set (Table 11), **SATURN**-1.5B improves average pass@3 from 10.1 to 24.2, while **SATURN**-7B improves from 36.1 to 64.2. On average, these models achieve pass@3 improvements of +14.0 and +28.1 respectively, confirming that **SATURN** effectively enhances LLM reasoning across both seen and unseen SAT difficulties.

## 3.4 RQ2: SATURN Demonstrates Strong Generalization to Math and Programming

We assess whether the reasoning capability learned by **SATURN** generalizes to math and programming tasks. We evaluate **SATURN**-1.5B and **SATURN**-7B on a range of reasoning benchmarks. The results shown in Table 3 provide a detailed comparison.

Table 3: Performance comparison on math and programming Benchmarks

| Model | AIME 24/25 | AMC 22/23 | Math500 | GPQA-D | LiveCodeBench | Avg. |
|---|---|---|---|---|---|---|
| GPT-4o (Aug'24) | 11.7 | - | 79.5 | 52.1 | 31.7 | - |
| Claude 3.5 Sonnet (Oct '24) | 15.7 | - | 77.1 | 59.9 | 38.1 | - |
| s1.1-1.5B | 1.7 | 25.3 | 42.2 | 29.3 | 2.2 | 20.1 |
| Still-3-1.5B-Preview | 23.3 | 74.7 | 84.6 | 34.8 | 17.1 | 46.9 |
| DeepSeek-R1-Distill-Qwen-1.5B | 21.6 | 65.1 | 83.6 | 30.3 | 16.4 | 43.4 |
| + SFT | 25.0 | 68.7 | 82.0 | 34.3 | 14.6 | 44.9 |
| **SATURN**-1.5B | **28.3** | **73.5** | **84.6** | **37.4** | **17.4** | **48.2** |
| z1-7B | 8.3 | 39.8 | 74.2 | 35.4 | 19.3 | 35.4 |
| s1.1-7B | 21.7 | 61.4 | 80.8 | 43.4 | 12.8 | 44.0 |
| OpenThinker-7B | 26.7 | 53.0 | 88.6 | 42.9 | 21.5 | 46.5 |
| DeepSeek-R1-Distill-Qwen-7B | **50.0** | 80.7 | 93.2 | 49.0 | 35.4 | 61.7 |
| **SATURN**-7B | 48.3 | **85.5** | **95.0** | **50.5** | **37.7** | **63.4** |

❶ **SATURN shows strong generalization to math and programming tasks.** On the AIME 24/25 benchmark, **SATURN**-1.5B outperforms z1-7B by 8.3 and s1.1-7B by 21.7. Similarly, **SATURN**-7B achieves a strong improvement on the Math500 dataset, increasing from 93.2 to 95.0. On LiveCodeBench, it improves from 35.4 to 37.7. On average, **SATURN**-1.5B improves by +4.9, and **SATURN**-7B improves by +1.8 across these benchmarks. These results highlight that **SATURN** enhances the reasoning performance of LLMs across various math and programming tasks, demonstrating strong generalization of the learned reasoning capabilities from SAT.

❷ **SATURN outperforms SFT on broader benchmarks.** Consistent with the observations in *SFT Memorizes, RL Generalizes* [8], SFT improves performance on math-focused benchmarks (AIME, AMC, and Math500) that are similar to its supervised training domain. However, on LiveCodeBench, SFT drops from 16.4 to 14.6, exhibiting an *alignment tax* [33], where specializing on a narrow domain compromises performance on other tasks. In contrast, **SATURN** improves performance across all benchmarks, with **SATURN**-1.5B reaching 17.4 on LiveCodeBench. Averaging across all

benchmarks, **SATURN**-1.5B outperforms the SFT counterpart by 3.3, demonstrating that **SATURN** generalizes effectively.

### 3.5 RQ3: SATURN Serves as a Complement and Further Enhances LLM Reasoning

RQ3 studies the relationship between **SATURN** and existing RL tasks. Beyond the DeepSeek-R1-Distill-Qwen-7B, we introduce two additional models: Qwen2.5-7B-Instruct-1M [39, 49] following Logic-RL [47] settings, and DeepScaleR-1.5B-Preview [29], which is further RL trained on 40k math and programming examples from DeepSeek-R1-Distill-Qwen-7B. We compare **SATURN** against several prior constructing RL task approaches, including Logic-RL [47], SPGA [7], and ScaleQuest [12], which represent strong baselines. Each approach is applied to different models for comparison. Results are summarized in Table 4.

Table 4: Comparison of **SATURN** and prior approaches across various LLMs.

| Model | AIME 24/25 | AMC 22/23 | Math500 | GPQA-D | LiveCodeBench | Avg. |
|---|---|---|---|---|---|---|
| SPGA-3 (82k) | 0.0 | 3.6 | 7.2 | 24.7 | 0.0 | 7.1 |
| ScaleQuest (25k) | 6.7 | 45.8 | 74.6 | 31.3 | 7.9 | 33.3 |
| Qwen2.5-7B-Instruct-1M | 5.0 | 41.0 | 74.4 | 32.3 | 9.8 | 32.5 |
| + Logic-RL (5k) | 6.7 | **49.4** | 72.0 | 29.3 | 9.0 | 33.3 |
| + Saturn (1k) | **10.0** | 47.0 | **74.8** | **37.9** | **11.3** | **36.2** |
| DeepSeek-R1-Distill-Qwen-7B | **50.0** | 80.7 | 93.2 | 49.0 | 35.4 | 61.7 |
| + Logic-RL (5k) | **50.0** | 80.7 | 93.4 | **52.0** | 35.7 | 62.4 |
| + Saturn (1k) | 48.3 | **85.5** | **95.0** | 50.5 | **37.7** | **63.4** |
| DeepScaleR-1.5B-Preview | 30.0 | 74.7 | 87.8 | 37.4 | 19.8 | 49.9 |
| + Logic-RL (5k) | 28.3 | **77.1** | 86.4 | 35.9 | 20.7 | 49.7 |
| + Saturn (0.5k) | **35.0** | 73.5 | **88.6** | **43.4** | **21.0** | **52.3** |

❶ **SATURN serves as a strong complement to math and programming.** On DeepScaleR-1.5B-Preview—despite being further RL trained with 40k math and programming examples—**SATURN** still brings additional improvements, raising the average score from 49.9 to 52.3. Notably, it improves AIME by +5.0 and GPQA-D by +6.0. ❷ **SATURN outperforms prior constructing RL task approaches across multiple models.** On Qwen2.5-7B-Instruct-1M, **SATURN** uses only 1k training examples but improves the average score from 32.5 to 36.2, achieving a relative improvement of +8.8% over Logic-RL trained with 5k examples. These results indicate that **SATURN** not only complements math and programming tasks, but also provides greater improvements compared to other constructing RL task approaches.

### 3.6 RQ4: SATURN Enhances Self-verification in LLMs' Reasoning Trajectories

RQ4 investigates whether **SATURN** affects LLMs' reasoning behavior. On Qwen2.5-7B-Instruct-1M, we observe a gradual increase in response length during training, as illustrated in Figure 4, replicating the lengthening phenomenon reported in the R1 and Logic-RL [11, 47].

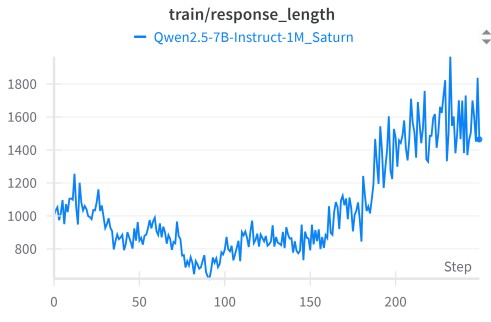

Figure 4: Response length trend during **SATURN** training on Qwen2.5-7B-Instruct-1M.

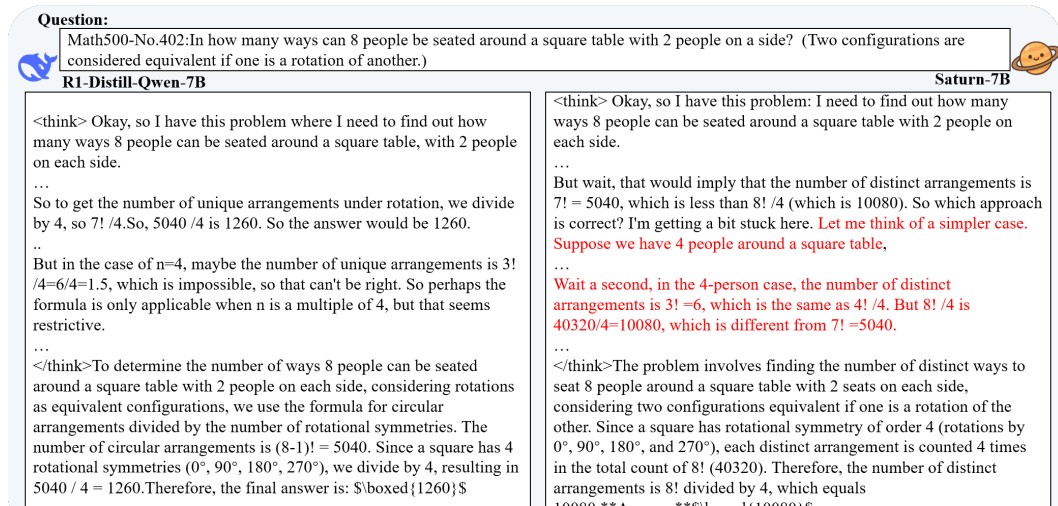

Figure 5: Comparison of reasoning trajectories on Math 500 Question No.402 between two models.

To examine whether such reasoning patterns generalize, we present case studies across SAT and math domains. Figure 12 shows that solving SAT variables requires rechecking all clauses, naturally encouraging self-verification. In Figure 5, **SATURN**-7B verifies intermediate conclusions within a small scenario and successfully chooses the correct solution path. In contrast, the baseline model reaches a wrong answer and skips verification, even when inconsistencies are detected.

Recent studies [15, 20] identify core behaviors shared by expert human reasoners and LLMs, such as **verification** and **backtracking**. These behaviors are domain-agnostic and provide fundamental reasoning patterns applicable to a wide range of tasks. In line with these findings, **SATURN** reinforces similar behaviors during SAT solving, leading to more structured reasoning trajectories. More reasoning trajectories are provided in Appendix J to illustrate how SATURN works. These results suggest that the self-verification patterns learned from SAT transfer well to math and programming tasks, improving reasoning robustness and reliability.

# 4 Discussion

## 4.1 Limitations of Reasoning Capability Learned from SATURN

During curriculum learning, we observed that as the number of training iterations increases, the improvements in math and programming tasks tend to plateau, which is consistent with the findings in Logic-RL. Detailed evaluation results are provided in Table 5. This plateau may stem from several factors: ❶ **Knowledge limitations. SATURN** improves formal logical reasoning but does not provide domain-specific knowledge supervision. This limits its effectiveness in tasks requiring mathematical or algorithmic knowledge. ❷ **Context window bottlenecks.** SAT problems are NP-complete tasks, and the required reasoning length grows exponentially with increasing problem difficulty. This leads to bottlenecks in the model's capability to handle increasingly complex tasks. ❸ **Limited plasticity and forgetting.** Model plasticity and catastrophic forgetting are known limitations that hinder further improvements with additional training stages [3, 13].

## 4.2 Potential of SATURN on Stronger Models

To explore the potential of **SATURN** to stronger models, we evaluate frontier LLMs on SAT tasks using the extended SAT instances. Results are shown in Appendix I. Although these LLMs exhibit stronger performance, they still make common errors such as hallucinating clauses, confidently committing to incorrect decisions, or failing to apply basic logical rules. Even more advanced LLMs still struggle to solve complex SAT problems. We believe **SATURN** remains a promising approach for enhancing reasoning in stronger LLMs. With sufficient computation, **SATURN** can offer a scalable, verifiable, and controllable path to further improve reasoning capabilities.

Table 5: Performance of multi-Stage **SATURN** iterations on math and programming benchmarks

| Model | AIME 24/25 | AMC 22/23 | Math500 | GPQA-D | LiveCodeBench | Avg. |
|---|---|---|---|---|---|---|
| DeepSeek-R1-Distill-Qwen-1.5B | 21.6 | 65.1 | 83.6 | 30.3 | 16.4 | 43.4 |
| **SATURN**-1.5B-Iteration-1 | 26.7 | 68.6 | 85.0 | 33.3 | 16.9 | 46.1 |
| **SATURN**-1.5B-Iteration-2 | 28.3 | **73.5** | 84.6 | **37.4** | **17.4** | **48.2** |
| **SATURN**-1.5B-Iteration-3 | **28.3** | 66.3 | **85.8** | 36.9 | 16.7 | 46.9 |
| DeepSeek-R1-Distill-7B | **50.0** | 80.7 | 93.2 | 49.0 | 35.4 | 61.7 |
| **SATURN**-7B-Iteration-1 | 48.3 | 83.1 | 94.6 | 50.5 | 36.6 | 62.6 |
| **SATURN**-7B-Iteration-2 | 48.3 | 85.5 | **95.0** | 50.5 | 37.7 | 63.4 |
| **SATURN**-7B-Iteration-3 | 46.7 | **87.9** | 93.2 | **58.1** | **38.1** | **64.8** |

## 5 Related Work

### 5.1 Constructing Reasoning Tasks for Reinforcement Learning

Several works have explored constructing reasoning tasks to improve the reasoning capability of LLMs. Logic-RL [47] and LMRL Gym [4] train LLMs on natural language logic puzzles but lack scalability due to their limited puzzle set. ScaleQuest [12], Entity-Deducing Game [52], and K&K [46] propose automatic generation of constructing questions, but rely on LLM sampling or handcrafted templates, making large-scale generation costly and hard to verify. CodeDPO [51] and PuzzBench [41] employ LLM-based verification, which may fail silently and cannot ensure correctness. Wolf Game [45, 48] focus on multi-step logic reasoning but offer no control over task difficulty, limiting their support for curriculum learning. Overall, these tasks fall short in scalability, verifiability, or controllable difficulty. See Appendix K for detailed comparisons.

### 5.2 SAT-Based Evaluation of LLM Reasoning Capability

Recent studies have evaluated the reasoning capability of LLMs on SAT problems. Most of these works focus on analyzing model behavior around the SAT phase transition [18, 30, 34], where problem hardness peaks. However, the phase transition theory is originally designed for heuristic SAT solvers and does not align well with the reflective and verification-based reasoning processes of humans or LLMs. These studies also lack a fine-grained scalable difficulty framework and typically divide difficulty based on the phase transition threshold. They are further limited to supervised fine-tuning and do not consider large reasoning models with long-CoT reasoning capability trained via RL. Our work addresses these limitations by building a progressive evaluation and curriculum learning pipeline, enabling precise difficulty control and the generalization of LLMs.

## 6 Conclusion and Future Work

We present **SATURN**, a SAT-based RL framework for unleashing and evaluating the reasoning capability of LLMs. By leveraging SAT's **scalability**, **verifiability**, and **controllable difficulty**, **SATURN** addresses key limitations of existing RL tasks. It constructs a multi-stage curriculum to gradually enhance reasoning, and introduces the **SATURN-2.6k** benchmark for controlled evaluation. Applied to DeepSeek-R1-Distill-Qwen, **SATURN** produces **SATURN**-1.5B and **SATURN**-7B, which show strong gains on unseen SAT tasks and generalize well to math and programming benchmarks.

In future work, we plan to: (1) apply **SATURN** to larger-scale LLMs, (2) break the existing paradigm's reliance on human-annotated data and explore new paths toward building LLMs with continuous self-evolution capabilities.

## Acknowledgements

This research is supported by the National Key R&D Program under Grant No. 2023YFB4503801, the National Natural Science Foundation of China under Grant No. 62192733, 62192730, 62192731, the Major Program (JD) of Hubei Province (No.2023BAA024), and the Beijing Natural Science Foundation under Grant No. 4264107.

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

# Appendix

## Table of Contents

## A   Pseudocode of SATURN Algorithm and Hyperparameters

Algorithm 1 presents the complete algorithmic workflow of **SATURN**.

---

**Algorithm 1** SATURN Learning_Loop$(n, k, l, \pi_\theta)$    # LLM represents $\pi_\theta$

```python
def Increase_difficulty(n, k, l, step=1):
    """
    Increments by D_step to increase SAT difficulty.
    """
    return n, k, l + D_step

def SATURN_learning_loop(n, k, l, LLM):
    for t in range(2):  # Max total curriculum iterations
        # Step 1: Curriculum Estimation Loop
        # Generate validation set
        Val_set = SAT_Construction(n, k, l, Val_size)
        pass_at_1 = evaluate(Val_set, LLM)  # Evaluate pass@1

        if pass_at_1 >= epsilon:
            n, k, l = Increase_difficulty(n, k, l)
        else:
            # Step 2: LLM Training Loop (only if pass_at_1 < epsilon)
            for i in range(10):  # Max GRPO steps per level
                # Generate training set
                D_train = SAT_Construction(n, k, l, Train_size)
                LLM = GRPO(LLM, D_train)

                # Re-generate validation set
                Val_set = SAT_Construction(n, k, l, Val_size)
                pass_at_1 = evaluate(Val_set, LLM)  # Evaluate pass@1
                if pass_at_1 >= epsilon:
                    break

    return LLM
```

---

Table 6 shows all **SATURN** hyperparameters.

Table 6: **SATURN** Hyperparameters

| Parameter | SATURN-1.5B | SATURN-7B |
|---|---|---|
| Initial $(n, k, l)$ | $(3, 5, 5)$ | $(3, 5, 13)$ |
| pass@k threshold $\epsilon$ | 0.5 | 0.75 |
| Training set size per step (`Train_size`) | 250 | 250 |
| Validation set size per step (`Val_size`) | 40 | 40 |
| Difficulty increment (`D_step`) | 1 | 2 |
| Curriculum iterations | 2 | 2 |
| Max GRPO steps per level | 10 | 10 |

## B    Construction Procedure of SAT Instances

This appendix describes the implementation details of the `SAT_Construction` algorithm introduced in Section 2.1. The goal is to generate $m$ satisfiable $(n, k, l)$-SAT instances in conjunctive normal form (CNF), each constructed to be consistent with a known Boolean solution. The algorithm ensures diversity and uniformity across sampled instances.

The construction procedure is outlined in Algorithm 2 and consists of the following steps:

1. A Boolean solution is randomly generated for the $k$ variables.
2. The constructor randomly selects $n$ variables from the $k$ variables to form a clause.
3. Under the constraint of satisfying the solution, the $n$ variables are randomly negated, resulting in $2^n - 1$ satisfiable clauses per variable set. From these steps, we uniformly obtain a large number of single-clause samples from the total of $2^k \cdot \binom{k}{n} \cdot (2^n - 1)$ valid (`solution`, `clause`) pairs, with an upper bound set to 100,000.
4. These (`solution`, `clause`) pairs are grouped into clusters based on the same solution.
5. Within each cluster, we randomly select $l$ clauses and shuffle their order to construct a full SAT instance that satisfies the corresponding solution.
6. Finally, we uniformly sample across clusters and remove duplicates to obtain a total of $m$ diverse SAT instances.

## C    Derivation and Validity of Difficulty Estimation for SAT Tasks on LLMs

As stated in Section 2.2, we propose a composite difficulty function Eq. (2) to estimate the reasoning difficulty of SAT problems for LLMs. This difficulty score combines a sparsity-based estimate of solution density with a structural complexity adjustment.

**Step 1: Sparsity-Based Estimate ($D_1$)**

We first estimate task difficulty by measuring the ratio between symbolic search cost and expected solution space size.

The symbolic search cost is approximately proportional to the number of variable-symbol combinations across decoding steps. For a problem with $k$ variables and $l$ decoding steps, we estimate:

$$\textbf{Search Cost} \propto k \cdot l \tag{4}$$

For Boolean constraint problems like SAT, the number of satisfying assignments is sparse. Assuming a random clauses, the expected number of valid solutions is:

$$\textbf{Expected Solutions} \approx 2^n - 1 \approx 2^n \tag{5}$$

where we approximate $2^n - 1$ by $2^n$ for analytical simplicity.

Taking the ratio and applying a logarithmic transform yields:

$$D_1(n, k, l) = \log_2(k \cdot l) - \log_2(2^n) = \log_2(k) + \log_2(l) - n \tag{6}$$

**Algorithm 2** `SAT_Construction`$(n, k, l, m)$

```
def SAT_Construction(n, k, l, m):
    V = [x_1, x_2, ..., x_k]   # k Boolean variables
    P = set()                  # pool of (solution, clause) pairs

    # Step 1-3: Generate (solution, clause) pairs
    while len(P) < 100000: # max number of single-clause samples
        vars = sample_variables(V, n)     # select n vars from V
                                          # e.g., [x_2, x_4, ...]
        solution = random_assign(vars)    # assign 0/1 to vars
                                          # e.g., {x_2:1, x_4:0, ...}
        clause = generate(vars, solution) # create clause
                                          # e.g., {x_2 or !x_4 ...}
        P.add((solution, clause))         # store pair

    # Step 4: Group by solution
    clusters = group_by_solution(P)  # solution -> list of clauses

    # Step 5-6: Construct m SAT instances
    instances = set()
    while len(instances) < m:
        solution, clauses = sample_cluster(clusters)  # select a group
        if len(clauses) < l:
            continue
        selected_clauses = sample_clauses(clauses, l) # pick l clauses
        shuffle(selected_clauses)
        instances.add((solution, selected_clauses))

    return instances
```

### Step 2: Structural Complexity Adjustment ($D_2$)

In addition to solution sparsity, symbolic reasoning difficulty also depends on the structural properties of the input formula. We construct a structure-aware term $D_2(n, k, l)$ based on two contributing factors.

First, consider the symbolic reuse density: when $k$ variables are reused across $n$ clauses, each variable is, on average, involved in $n/k$ constraints. This increases the interdependency between clauses. Since higher reuse leads to greater symbolic entanglement, making factorization more challenging for the model, we define the inverse ratio $\frac{k}{n}$ as a proxy for the structural cost:

$$\textbf{Variable Interaction Cost} = \frac{k}{n} \tag{7}$$

Second, the clause length $l$ determines the number of symbols that each clause contains. Longer clauses introduce more intra-clause dependencies, increasing local reasoning complexity. We approximate this with:

$$\textbf{Clause Width Cost} = \log_2(l) \tag{8}$$

These two components affect reasoning difficulty independently—one globally through variable sharing, and the other locally through clause complexity. We, therefore, combine them additively into:

$$D_2(n, k, l) = \frac{k}{n} + \log_2(l) \tag{9}$$

This additive form is justified as both terms grow monotonically with symbolic complexity and are approximately aligned in scale, enabling stable and interpretable difficulty estimation.

**Step 3: Final Composite Metric**

Combining both components yields:

$$D(n, k, l) = D_1(n, k, l) + D_2(n, k, l) \tag{10}$$

$$= \log_2(k) + \log_2(l) - n + \frac{k}{n} + \log_2(l) \tag{11}$$

$$= \log_2(k) + 2\log_2(l) - n + \frac{k}{n} \tag{12}$$

This final composite metric provides a stable and interpretable approximation of symbolic difficulty for LLMs, taking into account both sparsity and structure.

**Step 4: Ablation Study about estimation metric**

To further validate the necessity and effectiveness of Eq. (12) composite metric, we conduct an ablation study comparing alternative formulations. Table 7 reports the $R^2$ correlation of each metric with LLM performance (`pass@3`) across multiple model scales.

Table 7: Metric comparison across models.

| Metric Formula | R1-Qwen-1.5B | Saturn-1.5B | R1-Qwen-7B | Saturn-7B | Avg. | Std. Dev. |
|---|---|---|---|---|---|---|
| $-k - l \cdot \log_2(1 - \frac{1}{2^n})$ | 0.507 | 0.537 | 0.132 | 0.000 | 0.294 | 0.269 |
| $\alpha \cdot \frac{k}{n} + \beta \cdot \log_2(l),\ \alpha = 2,\ \beta = 1$ | 0.428 | 0.478 | 0.568 | 0.508 | 0.496 | 0.059 |
| $\alpha \cdot \frac{k}{n} + \beta \cdot \log_2(l),\ \alpha = 1,\ \beta = 1$ | 0.240 | 0.279 | 0.719 | 0.826 | 0.516 | 0.300 |
| $\log_2(k \cdot l) - \log_2(2^n - 1)$ | 0.875 | 0.893 | 0.451 | 0.157 | 0.594 | 0.356 |
| $\log_2(k) + 2 \cdot \log_2(l) - n + \frac{k}{n}$ | 0.707 | 0.746 | 0.724 | 0.501 | **0.670** | 0.113 |

Simpler metrics that consider only sparsity or only structural complexity perform worse overall, confirming that both components are essential for accurately capturing task difficulty. Our metric $\log_2(k) + 2 \cdot \log_2(l) - n + \frac{k}{n}$ combines both solution sparsity and structural complexity, which achieves the best overall correlation. The ratio between the solution space and the LLM's search space is the most crucial aspect. On SATURN-7B, the $R^2$ value is about 0.5 because the LLM already achieves over 90% accuracy on easy problems, which limits the observable linear correlation in that range. The corresponding figures are shown in Figures 10 and 11.

## D  Training Schedule for SATURN-1.5B and SATURN-7B

This appendix provides additional details on the **SATURN**-1.5B and **SATURN**-7B training. We conduct all experiments on 8 NVIDIA A100 40GB GPUs. We use the `OpenRLHF` framework[2] for GRPO training. The framework is designed to make RL training simple and user-friendly, which works well in our experiments. The hyperparameters used in training are summarized in Table 8. All other parameters not listed above consistently follow the default settings of `OpenRLHF`.

Figure 6 and Figure 7 illustrate the training curves of **SATURN**-1.5B and **SATURN**-7B. Both models show a clear upward trend in the average reward during training, accompanied by early fluctuations that gradually stabilize. The maximum reward curves quickly reach high values and maintain them throughout most of the training process. These results indicate that both **SATURN**-1.5B and **SATURN**-7B successfully learn to generate high-reward outputs, demonstrating effective SAT reward-guided optimization.

## E  Evaluation Hyperparameters

This appendix provides additional details on the hyperparameters used in the evaluation. We use the Hugging Face `lighteval` library[3] for math and programming evaluations. It offers efficient benchmarks, helping us assess LLMs' performance across various tasks while ensuring both computational

---

[2]`https://github.com/OpenRLHF/OpenRLHF`
[3]`https://github.com/huggingface/lighteval`

Table 8: `OpenRLHF` Training Hyperparameters

| Parameter | SATURN-1.5B | SATURN-7B |
|---|---|---|
| Actor learning rate | $5 \times 10^{-7}$ | $5 \times 10^{-7}$ |
| Initial KL coefficient | $1 \times 10^{-3}$ | $1 \times 10^{-3}$ |
| Batch size (train) | 2 | 2 |
| Batch size (rollout) | 2 | 2 |
| Samples per prompt | 8 | 8 |
| Prompt length (max) | 1024 | 1024 |
| Generation length (max) | 10000 | 8192 |
| Temperature | 0.8 | 1.0 |
| Zero redundancy stage | 3 | 3 |
| Use bf16 | Yes | Yes |
| KL estimator | k3 | k3 |
| Advantage estimator | GroupNorm | GroupNorm |
| Use reward normalization | Yes | Yes |

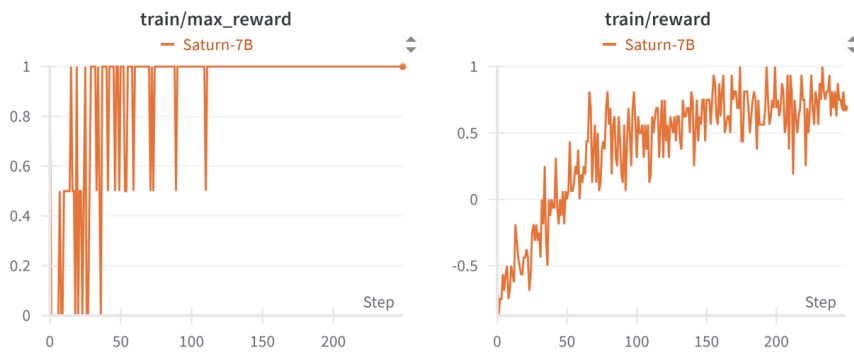

Figure 6: Training curves of various metrics for **SATURN**-1.5B.

efficiency and high-quality results. For the evaluation of the DeepSeek-R1 series distillation models, we use `lighteval` with the `Hugging Face-Open-R1` framework[4]. This framework effectively reproduces the evaluation results of the R1 series models.

During the evaluation process, we follow the parameter settings from `Hugging Face-Open-R1`, as shown in Table 9. Additionally, for LiveCodeBench, we also select the default v4_v5 version of

---

[4]`https://github.com/huggingface/open-r1`

Figure 7: Training curves of various metrics for **SATURN**-7B.

this framework. Due to the long inference budget required by LiveCodeBench, we set the maximum response length to 16K and generate four samples per instance to estimate pass@1. All other parameters not listed and mentioned above consistently follow the default settings of `Hugging Face-Open-R1`.

For larger closed-source models, we report the benchmark results from a public benchmark website[5].

Table 9: Evaluation Hyperparameters for `Hugging Face-Open-R1`

| Hyperparameter | Setting |
|---|---|
| Data type | `bfloat16` |
| Maximum model length | 32,768 |
| Maximum new tokens | 32,768 |
| Temperature | 0.6 |
| Top-$p$ (nucleus sampling) | 0.95 |

## F   Prompt Templates

This appendix provides the prompt templates used for evaluation, ensuring consistency and reproducibility across tasks. Figure 8 presents the format for SAT problem training and evaluation, while Figure 9 shows the template used for math, programming, and GPQA Diamond tasks.

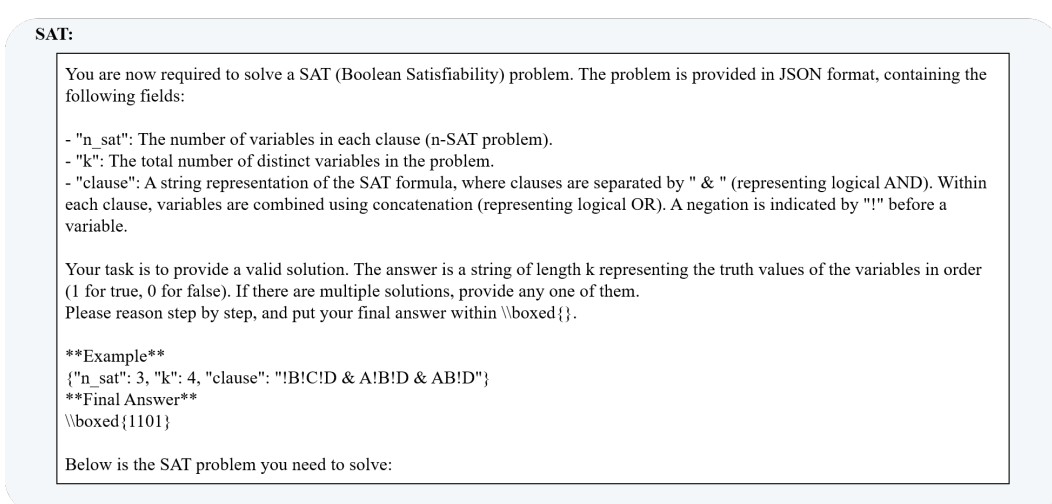

Figure 8: Prompt format used for SAT problem training and evaluation.

## G   Detailed Performances of SATURN models on SATURN-2.6k

This appendix provides additional details of **SATURN**-1.5B and 7B on **SATURN-2.6k** spanning 10 harder SAT difficulty levels. Experimental results are shown in Table 10–14, and Figures 10–11. We summarize two key observations:

❶ The pass@3 accuracy correlates strongly with the estimated SAT difficulty $D(n, k, l)$ across models. Specifically, the linear regression $R^2$ scores are 0.746 for **SATURN**-1.5B and 0.707 for DeepSeek-R1-Distill-Qwen-1.5B (Figure 10), and 0.5011 for **SATURN**-7B and 0.724 for DeepSeek-R1-Distill-Qwen-7B (Figure 11). These results indicate that our difficulty function effectively captures problem hardness, supporting the design of a curriculum learning schedule based on it. They also demonstrate that SAT is a reliable benchmark for evaluating reasoning capability.

---

[5]`https://artificialanalysis.ai/models`

Figure 9: Prompt format used for math, programming, and GPQA Diamond problems evaluation.

❷ Although our models are trained only on relatively easier SAT problems, they show consistent improvements on harder levels. As shown in Table 10–14, both **SATURN**-1.5B and **SATURN**-7B generalize well to more challenging problems, highlighting the effectiveness of our curriculum-driven training strategy.

Table 10: Full `pass@1` results on **SATURN-2.6k**

| Model | (3,7,40) | (3,5,25) | (3,5,20) | (3,6,20) | (3,7,20) | (4,7,40) | (4,8,40) | (4,7,20) | (6,7,40) | (5,8,40) | Avg. |
|---|---|---|---|---|---|---|---|---|---|---|---|
| R1-Distill-1.5B | 0.3 | 1.3 | 3.3 | 3.3 | 3.3 | 1.1 | 1.6 | 7.2 | 10.6 | 4.4 | 3.6 |
| **SATURN**-1.5B | **0.3** | **6.2** | **7.8** | **7.3** | **8.9** | **3.3** | **4.2** | **22.5** | **29.0** | **12.9** | **10.2** |
| R1-Distill-7B | 2.3 | 12.3 | 19.2 | 23.3 | 21.5 | 3.8 | 4.5 | 30.1 | 24.4 | 12.3 | 15.4 |
| **SATURN**-7B | **8.6** | **44.7** | **66.4** | **64.5** | **64.6** | **9.2** | **10.3** | **57.3** | **36.4** | **19.2** | **38.1** |

Table 11: Full `pass@3` results on **SATURN-2.6k**

| Model | (3,7,40) | (3,5,25) | (3,5,20) | (3,6,20) | (3,7,20) | (4,7,40) | (4,8,40) | (4,7,20) | (6,7,40) | (5,8,40) | Avg. |
|---|---|---|---|---|---|---|---|---|---|---|---|
| R1-Distill-1.5B | **1.0** | 3.8 | 9.4 | 9.1 | 9.3 | 3.3 | 4.7 | 19.3 | 28.9 | 12.5 | 10.1 |
| **SATURN**-1.5B | 0.8 | **15.8** | **19.5** | **18.5** | **21.6** | **9.5** | **11.3** | **50.3** | **62.5** | **31.7** | **24.2** |
| R1-Distill-7B | 6.5 | 31.2 | 45.6 | 52.8 | 49.9 | 10.4 | 11.8 | 65.6 | 56.2 | 31.3 | 36.1 |
| **SATURN**-7B | **22.9** | **78.2** | **94.1** | **93.5** | **93.3** | **24.3** | **26.5** | **91.0** | **72.4** | **45.7** | **64.2** |

Table 12: Full `pass@5` results on **SATURN-2.6k**

| Model | (3,7,40) | (3,5,25) | (3,5,20) | (3,6,20) | (3,7,20) | (4,7,40) | (4,8,40) | (4,7,20) | (6,7,40) | (5,8,40) | Avg. |
|---|---|---|---|---|---|---|---|---|---|---|---|
| R1-Distill-1.5B | **1.7** | 6.1 | 14.9 | 14.2 | 14.4 | 5.4 | 7.6 | 29.2 | 43.7 | 19.6 | 15.7 |
| **SATURN**-1.5B | 1.3 | **23.5** | **27.9** | **26.7** | **30.6** | **14.9** | **17.2** | **67.0** | **79.0** | **44.1** | **33.2** |
| R1-Distill-7B | 10.5 | 44.7 | 62.3 | 69.2 | 66.9 | 16.0 | 17.5 | 82.9 | 74.0 | 45.4 | 48.9 |
| **SATURN**-7B | **34.4** | **89.5** | **98.5** | **98.6** | **98.2** | **35.9** | **38.6** | **97.5** | **86.4** | **62.4** | **74.0** |

Table 13: Full `pass@7` results on **SATURN-2.6k**

| Model | (3,7,40) | (3,5,25) | (3,5,20) | (3,6,20) | (3,7,20) | (4,7,40) | (4,8,40) | (4,7,20) | (6,7,40) | (5,8,40) | Avg. |
|---|---|---|---|---|---|---|---|---|---|---|---|
| R1-Distill-1.5B | **2.3** | 8.1 | 19.9 | 18.9 | 18.9 | 7.6 | 10.1 | 37.4 | 55.7 | 25.8 | 20.5 |
| **SATURN**-1.5B | 1.8 | **30.1** | **34.2** | **32.8** | **37.5** | **19.8** | **22.1** | **78.2** | **87.8** | **52.5** | **39.7** |
| R1-Distill-7B | 14.3 | 54.6 | 73.6 | 78.7 | 77.8 | 20.9 | 22.2 | 91.6 | 84.3 | 56.3 | 57.4 |
| **SATURN**-7B | **43.7** | **94.2** | **99.6** | **99.7** | **99.5** | **45.0** | **47.6** | **99.2** | **92.2** | **73.5** | **79.4** |

Table 14: Full `pass@10` results on **SATURN-2.6k**

| Model | (3,7,40) | (3,5,25) | (3,5,20) | (3,6,20) | (3,7,20) | (4,7,40) | (4,8,40) | (4,7,20) | (6,7,40) | (5,8,40) | Avg. |
|---|---|---|---|---|---|---|---|---|---|---|---|
| R1-Distill-1.5B | **3.3** | 10.6 | 26.8 | 25.2 | 24.6 | 10.8 | 14.5 | 47.9 | 69.3 | 33.7 | 26.7 |
| **SATURN**-1.5B | 2.5 | **38.7** | **41.3** | **39.5** | **45.7** | **26.2** | **28.0** | **89.6** | **101.0** | **75.9** | **46.7** |
| R1-Distill-7B | 19.6 | 65.0 | 84.7 | 86.6 | 87.8 | 27.2 | 27.9 | 97.4 | 92.8 | 68.6 | 65.8 |
| **SATURN**-7B | **54.9** | **97.1** | **99.9** | **99.9** | **100.0** | **55.0** | **57.3** | **99.8** | 95.3 | **84.7** | **84.4** |

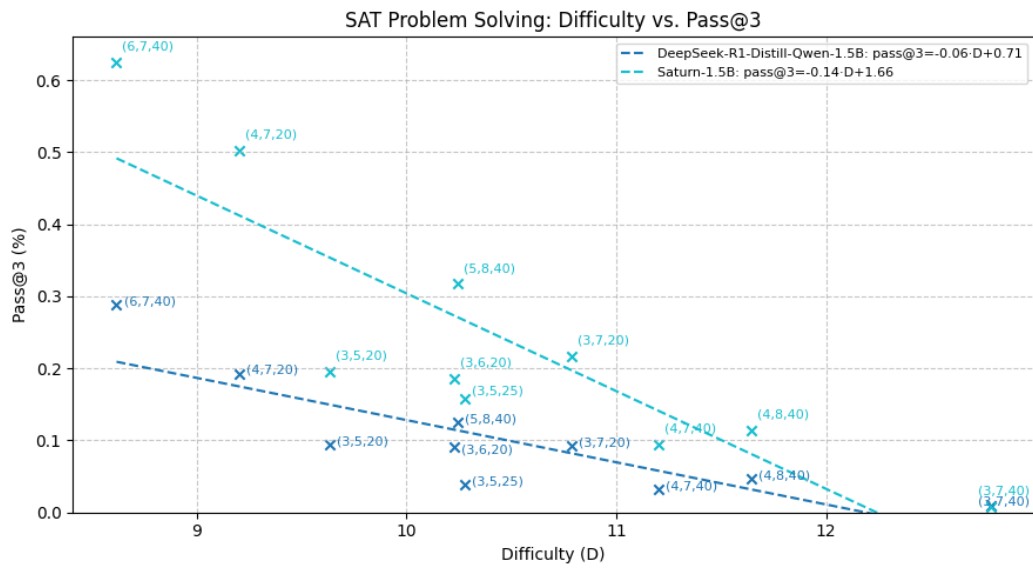

Figure 10: Scatter plots of `pass@3` versus estimated difficulty $D(n, k, l)$ for DeepSeek-R1-Distill-Qwen-1.5B and **SATURN**-1.5B, with linear regression fits. The linear regression for two models achieve $R^2$ values of 0.707 and 0.746 respectively.

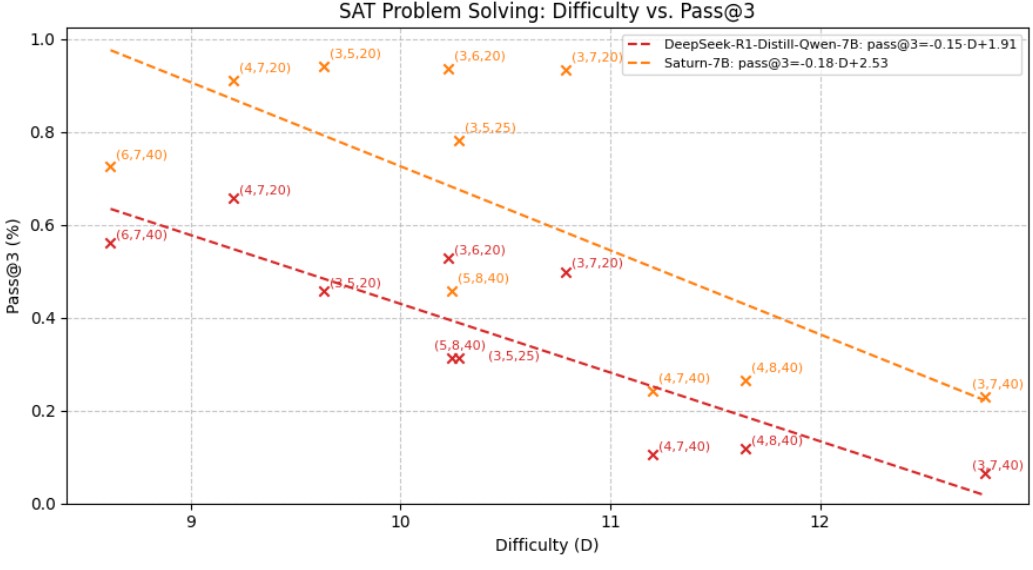

Figure 11: Scatter plots of `pass@3` versus estimated difficulty $D(n, k, l)$ for DeepSeek-R1-Distill-Qwen-7B and **SATURN**-7B, with linear regression fits. The linear regression for two models achieve $R^2$ values of 0.724 and 0.5011 respectively.

# H    Ablation Studies for SATURN

This appendix presents the ablation studies for **SATURN**, as shown in Table 15 and Table 16. Each training setting is denoted as $(n, k, l) \times$ `Train_size`, where $(n, k, l)$ controls SAT construction and `Train_size` is the number of training examples. Here, `Train_size` can also be written as `Train_size` $= D \times$ `num`, where $D$ is the number of difficulty levels and `num` is the number of samples per level in **SATURN-2.6k** (Figure 11 and 10). These experiments validate the effectiveness of curriculum learning and the design of various SAT training configurations.

Table 15: Ablation comparison on math and programming benchmarks

| Training Setting | AIME 24/25 | AMC 22/23 | Math500 | GPQA-D | LiveCodeBench | Avg. |
|---|---|---|---|---|---|---|
| $(3, 5, 10) \times 500$ | 45.0 | 84.3 | 93.2 | **53.5** | 35.1 | 62.2 |
| $(3, 5, 13) \times 500$ | 50.0 | 83.1 | 94.6 | 50.0 | 36.1 | 62.8 |
| $(3, 5, 15) \times 500$ | 35.0 | 68.7 | 86.6 | 46.0 | 31.3 | 53.5 |
| $(3, 5, 13) \times 1000$ | **51.7** | 81.9 | 94.0 | 47.5 | 37.2 | 62.5 |
| $((3, 5, 13) + (3, 5, 15)) \times 500$ (one epoch) | 43.3 | **86.7** | 93.0 | 49.5 | 35.8 | 61.7 |
| $(3, 5, 13) \times 500 + (3, 5, 15) \times 500$ | 48.3 | 85.5 | **95.0** | 50.5 | **37.7** | **63.4** |

Table 16: Ablation Study with Different Sampling Strategies and Training Budgets

| Training Setting | AIME 24/25 | AMC 22/23 | Math500 | GPQA-D | LiveCodeBench | Avg. |
|---|---|---|---|---|---|---|
| $(n, k, l) \times 100 \times 10$ + shuffle | 38.3 | 66.3 | 90.6 | 44.9 | 35.2 | 55.1 |
| $(n, k, l) \times 100 \times 10$ | 46.7 | 85.5 | 93.2 | 50.5 | 36.9 | 62.3 |
| $(n, k, l) \times 200 \times 5$ + shuffle | 48.3 | 81.9 | 93.0 | **52.0** | 35.8 | 62.2 |
| $(n, k, l) \times 200 \times 5$ | 46.7 | **88.0** | 93.2 | 48.0 | 35.8 | 62.6 |
| $(3, 5, 13) \times 500 + (3, 5, 15) \times 500$ | **48.3** | 85.5 | **95.0** | 50.5 | **37.7** | **63.4** |

In Table 15, we evaluate the impact of SAT difficulty, training budgets, and curriculum structure. We draw two key conclusions:

❶ **SATs that are too easy or too hard hinder model learning.** Training solely on easy $(3, 5, 10) \times 500$ or hard $(3, 5, 15) \times 500$ instances results in lower average scores (62.2 and 53.5, respectively). In contrast, moderate-difficulty SATs $(3, 5, 13) \times 500$ yield a higher score of 62.8, showing that balanced difficulty is essential for effective reasoning development.

❷ **Multi-stage curriculum learning outperforms flat or mixed training.** Curriculum learning with progressively increasing SAT difficulty $(3, 5, 13) \times 500 + (3, 5, 15) \times 500$ achieves the highest average score of 63.4. In contrast, one-epoch mixed training $((3, 5, 13) + (3, 5, 15)) \times 500$ only reaches 61.7, despite using the same total number of examples. Furthermore, simply scaling up a single-stage setting $(3, 5, 13) \times 1000$ yields 62.5, which is also inferior to the curriculum. These results indicate that progressive difficulty scheduling is more effective than either flat or mixed training with the same or larger data budget.

Table 16 further investigates the impact of training thresholds and step sizes under a fixed total training budget.

❸ **Gradual difficulty progression outperforms random shuffling of difficulty levels.** Both $(n, k, l) \times 100 \times 10$ and $(n, k, l) \times 200 \times 5$ perform better when difficulty levels follow a gradual progression (62.3 and 62.6), compared to random shuffling of difficulty levels (55.1 and 62.2). This demonstrates that a curriculum learning approach with progressive difficulty scheduling is more effective.

❹ **Excessively fine-grained difficulty levels hinder performance.** Training with overly fine-grained difficulty levels, such as $(n, k, l) \times 100 \times 10$, results in lower performance (55.1) compared to coarser steps like $(n, k, l) \times 200 \times 5$ (62.6). Both of these configurations perform worse than the two-stage curriculum $(3, 5, 13) \times 500 + (3, 5, 15) \times 500$, which achieves the highest performance with an average score of 63.4. This indicates that excessively fine-grained difficulty levels prevent the model from effectively mastering each level before moving on to the next, hindering overall learning.

# I  Behavior of Stronger LLMs on Extended SATURN Tasks

This appendix demonstrates the performance of stronger LLMs on more challenging SAT tasks. The experimental results are shown in Table 17. Even the strongest LLMs available today cannot solve complex SAT tasks as effectively as humans using simple reflection and verification search. Due to the long CoT involved, the full LLMs' outputs are provided in the supplementary material.

Table 17: One-shot performance of stronger LLMs on extended **SATURN** tasks. Models with ✔ successfully solve the corresponding difficulty of SAT tasks. Kimi-1.5 solves the task but with significantly longer reasoning chains.

| SAT Task (n, k, length) | GPT-4o | O1-mini | DeepSeek-V3 | R1 | Kimi-1.5 |
|---|---|---|---|---|---|
| (3, 5, 30) | ✗ | ✔ | ✗ | ✔ | ✔* |
| (4, 7, 80) | ✗ | ✗ | ✗ | ✗ | ✗ |

To provide a baseline comparison, we also tested a CDCL SAT solver [6] on the SATURN-2.6k test set. The results are as follows:

Table 18: CDCL SAT solver performance on SATURN-2.6k test set.

| Metric | Value |
|---|---|
| Total instances | 1000 |
| Satisfiable | 1000 |
| Unsatisfiable | 0 |
| Valid SAT Models | 1000 |
| Model Accuracy | 100.00% |
| Total time taken (s) | 0.14 |

Table 18 shows that the CDCL SAT solver significantly outperforms current LLMs like DeepSeek-R1, both in terms of runtime and accuracy. **SATURN** applied to today's strongest LLMs still has great potential.

# J  Examples of Different LLMs' Reasoning Trajectories

This appendix presents examples of different LLMs' reasoning trajectories, as shown in Figures 5 and 12. The full LLMs' outputs are provided in the supplementary material.

In the case of Math500-41 (Table 13), **SATURN** improves its ability to avoid unnecessary calculations and dead-end reasoning paths. By leveraging the self-verification patterns learned from SAT tasks, such as "I made a mistake earlier," **SATURN**-7B can better navigate through the problem, discarding invalid paths earlier in the process.

In the case of Math500-402 (Table 5), **SATURN**-7B demonstrates a stronger ability to self-verify intermediate conclusions in a smaller scenario (with 4 individuals). While DeepSeek-R1-Distill-Qwen-7B also tries to identify a smaller scenario, it fails to recheck the result when an inconsistency is found, instead stating, "perhaps I'm overcomplicating this." In contrast, **SATURN**-7B can identify the error and re-verify the results within this small scenario, ultimately selecting the correct solution from two possible candidates.

In conclusion, **SATURN**-7B exhibits enhanced self-verification capabilities. LLMs sometimes confidently claim that a wrong answer is correct. Solving SAT tasks inherently involves frequent and fine-grained clause verification, which trains LLMs to perform precise checking during reasoning. The self-verification patterns learned from SAT tasks help LLMs solve math problems more effectively by selecting correct solutions from multiple options. These results suggest that the self-verification mechanisms developed during SAT learning (Table 12) generalize well to math and programming tasks, improving the LLMs' reasoning robustness and reliability.

---

[6] https://en.wikipedia.org/wiki/Conflict-driven_clause_learning

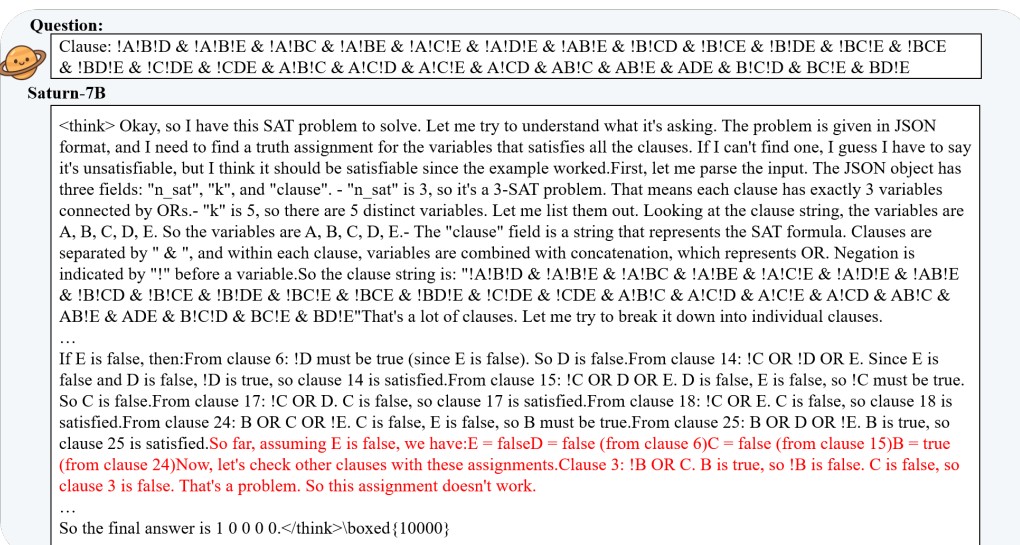

Figure 12: Case study of **SATURN**-7B's reasoning trajectory on a SAT instance.

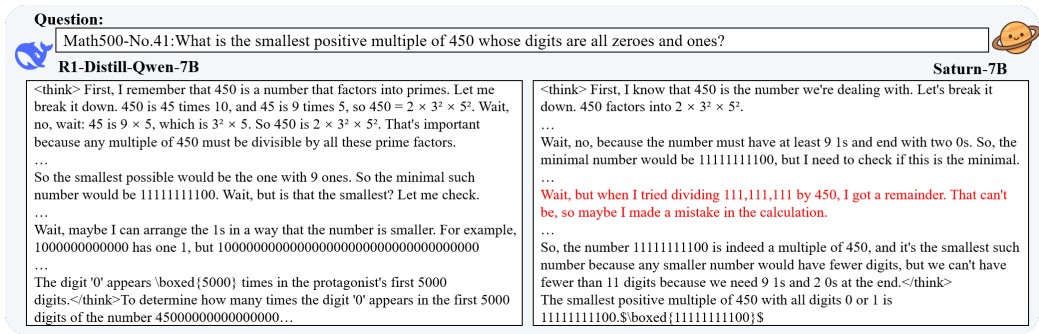

Figure 13: Comparison of reasoning trajectories on Math 500 Question No.41 between two models.

# K Detailed Comparison of Constructed Reasoning Tasks

This appendix provides a detailed comparison between *K&K logic puzzle* and *SAT problem.* Illustrative examples of each type are shown in Figure 14. While both are designed to test reasoning capability, they differ in construction cost, output format complexity, and difficulty control.

**Construction Cost.** K&K puzzles require translating symbolic logic into natural language. This involves paraphrasing logical constraints into grammatically and semantically valid sentences, which increases construction cost. In contrast, SAT problems are purely symbolic and follow a standard format. As shown in [18], symbolic SAT and its natural language version yield similar reasoning performance for LLMs, suggesting the symbolic form is sufficient.

**Output Format Complexity.** K&K puzzles require answers that list each character's identity in order, such as (1) A is a knight, (2) B is a knave. This format imposes strict requirements on structure, making it harder for LLMs to follow instructions. In practice, we observe that models struggle to learn this format in early training stages. SAT problems only require a fixed-length binary string wrapped in \boxed{}, which simplifies output and improves consistency during training.

**Difficulty Control.** K&K puzzles use the number of characters to control difficulty, which is coarse-grained. SAT problems allow fine-grained control via clause structure and variable interactions. We further define an estimation of SAT task difficulty for LLMs as $D(n, k, l) = \log_2(k) + 2\log_2(l) -$

| Logic-RL Knights and Knaves (K&K) logic puzzle | Boolean Satisfiability Problem |
|---|---|
| Prompt: You are a helpful assistant. The assistant first thinks about the reasoning process in the mind and then provides the user with the answer. The reasoning process and answer are enclosed within <think> </think> and<answer> </answer> tags, respectively, i.e., <think> reasoning process here </think><answer> answer here </answer>. Now the user asks you to solve a logical reasoning problem. After thinking, when you finally reach a conclusion, clearly state the identity of each character within <answer> </answer> tags. i.e., <answer> (1) Zoey is a knight
(2) ... </answer>.

A very special island is inhabited only by knights and knaves. Knights always tell the truth, and knaves always lie. You meet 7 inhabitants: Noah, Victoria, Ava, Lily, Daniel, Alexander, and Matthew. Noah said that Matthew is a knight and Victoria is a knight. "Lily is a knave if and only if Matthew is a knight," Victoria mentioned. In a statement by Ava: "Matthew is a knight or Victoria is a knight". Lily was heard saying, "If Victoria is a knave then Matthew is a knave". Daniel commented, "Alexander is not a knight". Alexander stated, "Noah is a knight or Ava is a knave". "Ava is a knave" - Matthew. So who is a knight and who is a knave? | Prompt: You are now required to solve a SAT (Boolean Satisfiability) problem. The problem is provided in JSON format, containing the following fields:

- "n_sat": The number of variables in each clause (n-SAT problem).
- "k": The total number of distinct variables in the problem.
- "clause": A string representation of the SAT formula, where clauses are separated by " & " (representing logical AND). Within each clause, variables are combined using concatenation (representing logical OR). A negation is indicated by "!" before a variable.

Your task is to provide a valid solution. The answer is a string of length k representing the truth values of the variables in order (1 for true, 0 for false). If there are multiple solutions, provide any one of them.
Please reason step by step, and put your final answer within \boxed{}.

**Example**
{"n_sat": 3, "k": 4, "clause": "!B!C!D & A!B!D & AB!D"}
**Final Answer**
\boxed{1101}

Below is the SAT problem you need to solve:
{"n_sat": 3, "k": 5, "clause": "!A!B!E & !A!BC & !A!DE & !AC!D & !ACE & !B!C!E & A!B!E & A!CE & ACD & BD!E"} |

Figure 14: Comparison of Knights and Knaves (K&K) logic puzzle and SAT problem.

$n + \frac{k}{n}$. Adding a clause to a SAT formula never decreases its difficulty, for both humans and LLMs. This makes SAT more suitable for curriculum learning.

In summary, while K&K puzzles provide linguistic diversity, SAT problems are more efficient in construction, output consistency, and difficulty regulation, making them preferable for training LLMs at scale.

# L  Word Cloud of SATURN-7B's Outputs on GPQA Diamond

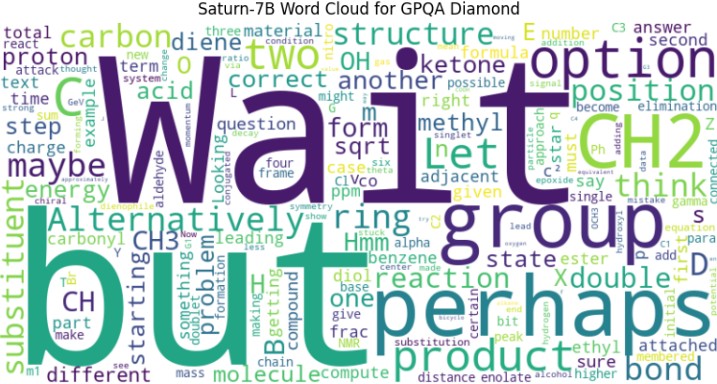

Figure 15: Word cloud of **SATURN**-7B's generated answers on GPQA Diamond. Frequently used tokens are shown in larger fonts.

Figure 15 shows the word cloud of **SATURN**-7B's generated answers on GPQA Diamond, highlighting its frequent use of self-verification patterns in reasoning.

