# OpenReview forum: "SATURN: SAT-based Reinforcement Learning to Unleash LLMs Reasoning"
_NeurIPS.cc/2025/Conference — NeurIPS 2025 spotlight_

### Official Review · Reviewer_1uh9 · 2025-06-24

**Clarity:** 3
**Significance:** 3
**Originality:** 2
**Rating:** 5
**Confidence:** 4

**Summary:**

The authors propose a reinforcement learning loop that trains large language models (LLMs) to solve SAT queries. The reward scheme is simple; the SAT query construction scheme, and especially the task difficulty estimation, is well-motivated and not obvious (in a good way). After training, DeepSeek-R1-Distill-Qwen-1.5B and 7B improve performance on SAT queries and, more surprisingly, on other tasks, like math and programming. The authors suggest that this generalization is due to improved self-verification capabilities of LLMs + SATURN.

**Questions:**

- Which aspects of the difficulty estimation metric are most crucial?
- (Optional) Is the author's metric good at estimating difficulty for CDCL SAT solvers?

**Ethical Concerns:**

["NO or VERY MINOR ethics concerns only"]

**Final Justification:**

The paper should be accepted because (1) the task difficulty estimation is well-motivated and works empirically; (2) the task construction method follows nicely from the difficulty estimation metric; and (3) SATURN's ability to generalize beyond SAT is interesting and significant.

I asked the authors how their metric compares to simpler alternatives and they provided interesting data in the rebuttal that improves the paper. I also asked them to clarify some details, like the "\boxed{}" syntax and the satisfiability of the queries, and they obliged. I am satisfied with their responses.

The main weakness I see is the originality of the work. For example, Logic-RL is similar. The main differentiating factor is that they can control the difficulty and they perform better than Logic-RL. I accept this argument.

**Limitations:**

yes

**Quality:**

3

**Strengths And Weaknesses:**

## Strengths
- The paper is clear, well-written, and well typeset.
- The task difficulty estimation is well-motivated and works empirically. The task construction method follows nicely from the difficulty estimation metric.
- The evaluation is solid.
- SATURN's ability to generalize beyond SAT is interesting and significant.

## Weaknesses
- Logic-RL is quite similar, although, of course, there is value in SATURN's data generation methodology.
- While the task difficulty estimation is well-motivated and empirically effective, the paper would benefit from an argument for why this metric is the best choice. Why does something simpler not work just as well? Are all parts of the estimation method necessary? A comparison to alternatives or an ablation study for the metric specifically may be appropriate, given that this is the paper's strongest contribution.
- The syntax constraint (e.g., "\boxed{}") is never really explained. It was not until the appendix that it became clear that the LLM must output a satisfying model and not just SAT/UNSAT.
- It is not obvious outside of the appendix that all queries are constructed to be satisfiable. The LLM is acting as a model finder, not as a SAT solver. This makes sense—you can avoid calling a SAT solver on all your generated queries to get the ground truth—but it should be explicit.

---

> ### Author Rebuttal · Authors · 2025-07-30
>
> Thank you for your valuable comments. We will address your concerns point by point.
>
> ## Q1: Which aspects of the difficulty estimation metric are most crucial?
>
> Thank you for your insightful and professional question. In our early exploration, we conducted an ablation study of the SAT difficulty estimation metric on our SATURN‑2.6K test set. The dataset contains 10 categories of problems with 100 instances each, totaling 1,000 problems (Appendix G).
>
> **📊 Selected SAT Problem Difficulty $(n, k, l)$**
> | $n$ | 3 | 3 | 3 | 3 | 3 | 4 | 4 | 4 | 6 | 5 |
> |:-----:|:-----:|:-----:|:-----:|:-----:|:-----:|:-----:|:-----:|:-----:|:-----:|:-----:|
> | **$k$** | **7** | **5** | **5** | **6** | **7** | **7** | **8** | **7** | **7** | **8** |
> | **$l$** | **40** | **25** | **20** | **20** | **20** | **40** | **40** | **20** | **40** | **40** |
>
> The pass@3 results are shown below:
>
> **📊  Full pass@3 results on SATURN-2.6k (Test Set)**
>
> | Model / $(n, k, l)$            | (3,7,40) | (4,7,40) | (4,8,40) | (5,8,40) | (6,7,40) | (3,5,25) | (4,7,20) | (3,7,20) | (3,6,20) | (3,5,20) | Avg.  |
> |------------------|----------|----------|----------|----------|----------|----------|----------|----------|----------|----------|-------|
> | R1-Distill-1.5B  | **1.0**  | 3.3      | 4.7      | 12.5     | 28.9     | 3.8      | 19.3     | 9.3      | 9.1      | 9.4      | 10.1  |
> | **SATURN-1.5B**  | 0.8      | **9.5**  | **11.3** | **31.7** | **62.5** | **15.8** | **50.3** | **21.6** | **18.5** | **19.5** | **24.2** |
> | R1-Distill-7B    | 6.5      | 10.4     | 11.8     | 31.3     | 56.2     | 31.2     | 65.6     | 49.9     | 52.8     | 45.6     | 36.1  |
> | **SATURN-7B**    | **22.9** | **24.3** | **26.5** | **45.7** | **72.4** | **78.2** | **91.0** | **93.3** | **93.5** | **94.1** | **64.2** |
>
> We then performed linear regression between each difficulty metric and model pass@3 (averaged across four models, with variance reported). The metrics capture different aspects including sparsity and structural complexity of SAT, as derived in Appendix C. The results are as follows:
>
> **📊 Metric Comparison Table**
>
> | Metric Formula                                                                       | R1-Qwen-1.5B | Saturn-1.5B | R1-Qwen-7B | Saturn-7B | Avg.   | Std. Dev. |
> | :----------------------------------------------------------------------------------: | :----------: | :---------: | :--------: | :-------: | :----: | :--------: |
> | $-k - l \cdot \log_2(1 - \frac{1}{2^n})$                                             |    0.507     |    0.537    |   0.132    |   0.000   | 0.294  |   0.269    |
> | $\alpha \cdot \frac{k}{n} + \beta \cdot \log_2(l)$, $\alpha{=}2$, $\beta{=}1$       |    0.428     |    0.478    |   0.568    |   0.508   | 0.496  |   0.059    |
> | $\alpha \cdot \frac{k}{n} + \beta \cdot \log_2(l)$, $\alpha{=}1$, $\beta{=}1$       |    0.240     |    0.279    |   0.719    |   0.826   | 0.516  |   0.300    |
> | $\log_2(k \cdot l) - \log_2(2^{n} - 1)$                                             |    0.875     |    0.893    |   0.451    |   0.157   | 0.594  |   0.356    |
> | $\log_2(k) + 2 \cdot \log_2(l) - n + \frac{k}{n}$                                    |    0.707     |    0.746    |   0.724    |   0.501   | **0.670** |   0.113    |
>
> Our metric $\log_2(k) + 2 \cdot \log_2(l) - n + \frac{k}{n}$ combines both solution sparsity and structural complexity, and achieves the best overall correlation. The ratio between the solution space and the LLM’s search space is the most crucial aspect. **On SATURN‑7B, the R² value is about 0.5 because pass@3 already achieves over 90% on easy problems**, which limits the observable linear correlation in that range. Due to rebuttal length constraints, the corresponding figures are shown in Appendix Figures 9 and 10.
>
> We will include this ablation study in future revisions to strengthen the paper. Thank you again for your valuable suggestion.
>
> ## Q2: (Optional) Is the author's metric good at estimating difficulty for CDCL SAT solvers?
>
> This is an excellent question. To fairly compare with LLM results, we tested a CDCL SAT solver on the SATURN-2.6k test set (see Appendix G). The results are as follows:
>
> ```python
>   Total instances:   1000
>   Satisfiable:       1000
>   Unsatisfiable:     0
>   Valid SAT Models:  1000
>   Model Accuracy:    100.00%
> Total time taken: 0.14 seconds
> ```
>
> It show that the CDCL SAT solver significantly outperforms current LLMs, even DeepSeek-R1.
>
> Next, we focused on 3-SAT problems with 100 variables, varying the number of clauses $l$ from 200 to 800 in steps of 5. We recorded the solving time across SAT instances of increasing clause numbers. Due to rebuttal format limits that forbid image uploads, we provide the solving times as a list, where the values represent solver runtimes:
>
> ```
> ['3.30e-05', '4.01e-05', '7.00e-05', '3.02e-04', '2.46e-03', '1.48e-03', '1.26e-03', '8.68e-04', '5.51e-04', '4.76e-04', '4.25e-04', '3.95e-04', '2.98e-05']
> ```
>
> Since the list is too long, for clarity we report the average of every 10 consecutive numbers. As a result, heuristic SAT solvers still rely on traditional phase transition theory as their difficulty metric. The difficulty estimation we present in the paper is specifically designed to simulate human-like and LLM-like search, verification, and backtracking processes in solving SAT. Our metric is based on the solution space and the search space of SAT problems.
>
> We also discuss this distinction in Section 2.3 of the paper: However, RSB theory is designed for heuristic SAT solvers. For humans or LLMs solving SAT problems through logical steps like trial, verification, and reasoning, such solver-like phase transitions are hardly observable in human-like thinking processes.
>
> ## Weakness 1: Logic-RL is quite similar, although, of course, there is value in SATURN's data generation methodology.
>
> Thank you for your comment. We agree that SATURN and Logic-RL share the same perspective about LLM reasoning. However, SATURN offers three key advantages: (1) **Controllable difficulty.** SAT-based generation with our estimator allows stable, fine-grained curriculum control. (2) **Better format following.** LLMs trained with SATURN is easier to follow the required output format after RL, and (3) Stronger results. Our main experiments in RQ3 show that, SATURN consistently outperforms Logic-RL on math and programming tasks. Detailed results can be found in Appendix L.
>
> ## Weakness 2: Why is this metric the best choice.
>
> Please see the response to **Q1**.
>
> ## Weakness 3: The syntax constraint (e.g., "\boxed{}") is unclear until the appendix.
>
> We apologize for the confusion. In the revised version, we have moved this key information from the appendix to the main text.
>
> ## Weakness 4: It is not obvious outside of the appendix that all queries are constructed to be satisfiable.
>
> Thank you for the thoughtful and professional suggestion. We will clarify in the main text that all generated SAT queries are satisfiable by construction. This design avoids calling a SAT solver for ground-truth. We appreciate the reviewer for pointing out the need to make this explicit.
>
> Finally, thank you very much for recognizing our work. Your suggestions are highly valuable to us. We sincerely appreciate your feedback!

---

> > ### Comment · Reviewer_1uh9 · 2025-08-03
> >
> > Thank you for engaging with the questions and making adjustments to the paper.
> > Including the ablation study, along with clarifications about \boxed and the
> > satisfiability of queries, in future revisions will strengthen the paper.
> >
> > It seems like we would need a more challenging benchmark set to answer Q2. But,
> > like I stated in the original review, answering this is not a top priority.

---

> ### Author Response · Authors · 2025-08-04
>
> Thank you for your follow-up comment. We would like to provide further clarification for **Q2**, including our benchmark selection and additional evaluation results:
>
> 1. **Benchmark choice.** Our original evaluation followed the widely adopted setup of DeepSeek-R1 [1], including AIME 2024/2025, AMC 2022/2023, Math500, GPQA-D, and LiveCodeBench, which are recognized as highly challenging benchmarks.
>
> 2. **Additional benchmarks.** Following [2,3], we further evaluate on GSM8K [4], Minerva Math [5], and OlympiadBench [6], which are also recognized as challenging benchmarks. The results are shown below:
>
> | Model             |  GSM8K  | Minerva Math | OlympiadBench |  Avg.  |
> |-------------------|:-------:|:------------:|:-------------:|:------:|
> | R1-Distill-1.5B      |  81.1   |    31.6      |    38.8       |  50.5  |
> | Saturn-1.5B       | **85.1**|  **34.2**    |   **41.0**    | **53.4**|
> | R1-Distill-7B        |  92.3   |    42.3      |    51.1       |  61.9  |
> | Saturn-7B         | **92.9**|  **43.8**    |   **51.6**    | **62.8**|
>
> Our method consistently improves over R1-Qwen on all new benchmarks, further demonstrating its effectiveness across challenging reasoning tasks.
>
> Finally, we appreciate your constructive feedback. Please feel free to contact us if you have any further questions.
>
> **References**
> [1] Guo, Daya, et al. *"Deepseek-r1: Incentivizing reasoning capability in llms via reinforcement learning."* arXiv preprint arXiv:2501.12948 (2025).
> [2] Hu, Jingcheng, et al. *"Open-reasoner-zero: An open source approach to scaling up reinforcement learning on the base model."* arXiv preprint arXiv:2503.24290 (2025).
> [3] Zeng, Weihao, et al. *"Simplerl-zoo: Investigating and taming zero reinforcement learning for open base models in the wild."* arXiv preprint arXiv:2503.18892 (2025).
> [4] Cobbe, Karl, et al. *"Training verifiers to solve math word problems."* arXiv preprint arXiv:2110.14168 (2021).
> [5] Lewkowycz, Aitor, et al. *"Solving quantitative reasoning problems with language models."* arXiv preprint arXiv:2206.14858 (2022).
> [6] He, Chaoqun, et al. *"Olympiadbench: A challenging benchmark for promoting agi with olympiad-level bilingual multimodal scientific problems."* arXiv preprint arXiv:2402.14008 (2024).

---

### Official Review · Reviewer_QZ5H · 2025-06-28

**Clarity:** 2
**Significance:** 2
**Originality:** 2
**Rating:** 4
**Confidence:** 4

**Summary:**

This paper proposes SATURN, a curriculum-based reinforcement learning (RL) framework that uses SAT problems to train and evaluate large language models (LLMs) on reasoning tasks.  A dataset named SATURN-2.6k is introduced to benchmark performance over varying difficulty levels. The authors apply their approach to LLMs like DeepSeek-R1-Distill-Qwen and show improvements on SAT solving tasks and modest improvements on math and programming benchmarks.

**Questions:**

1. How do you ensure that the SAT tasks used in SATURN actually require multi-step logical reasoning rather than simple token pattern recognition or template-based decoding?

2. Can you provide a comparative study of SATURN vs. supervised fine-tuning on the same SAT dataset to justify the use of reinforcement learning?

3. Is there any evidence that training on richer formal logic problems (e.g., QBF, theorem proving) would provide stronger reasoning benefits than SAT? If so, why was SAT selected despite its limited expressiveness?

4. How does the reasoning learned from SAT generalize to tasks involving semantic understanding or abstraction, which SAT does not cover?

5. Have you tested model behavior on harder SAT instances near the phase transition (e.g., α ≈ 4.2 for 3-SAT) with much more variables where symbolic reasoning becomes nontrivial?

**Ethical Concerns:**

["NO or VERY MINOR ethics concerns only"]

**Final Justification:**

The authors' response addressed my concerns regarding dataset simplicity, the suitability of SAT for reasoning, and the role of RL. They provided additional empirical evidence and clarified theoretical motivations. I now find the claims more substantiated and have raised my score to 4.

**Limitations:**

Yes

**Quality:**

2

**Strengths And Weaknesses:**

**Strengths**
1. The paper introduces SATURN, a curriculum-based reinforcement learning framework that leverages SAT problems to train and evaluate LLMs on complex reasoning tasks.
2. The authors provide a benchmark dataset, SATURN-2.6k, which enables evaluation across varying difficulty levels.
3. The models fine-tuned with SATURN show improvements on SAT tasks and some generalization to math and code domains.

**Weaknesses**
1. **Overly simple SAT datasets**:
- The SAT problems used in training (e.g., 3-SAT with clause-to-variable ratios far from the phase transition) are syntactically and semantically simple.
- Many tasks can likely be solved by pattern matching or shallow heuristics rather than logical reasoning, especially with SAT instances constructed to be trivially satisfiable.
- Despite the theoretical hardness of SAT, the practical instances used here do not reflect the complexity needed to truly test or build reasoning capabilities.

2. **Whether SAT is suitable for reasoning skills**:
- The paper claims that solving SAT via RL builds general reasoning skills. However, SAT solving is fundamentally search-based and benefits from symbolic constraint satisfaction strategies — not the kind of causal reasoning or abstraction often required in math or programming.

- The assumption that SAT instances naturally induce verification and self-reflection is speculative and lacks empirical justification or ablation analysis.

3. **Lack of RL Analysis**:
The contribution is based on the use of RL, but the role of RL is under-analyzed. There is no comparison to standard supervised fine-tuning on the same tasks, and it remains unclear whether RL offers any unique benefit over direct optimization on pass@k.

4. **Modest improvements** : Improvements on downstream tasks (math, code) are modest and possibly attributable to performance variance introduced by training or domain transfer from additional training data, not necessarily from enhanced reasoning from SAT tasks.

---

> ### Author Rebuttal · Authors · 2025-07-30
>
> ## Q1: How do you ensure that SAT tasks require multi-step reasoning?
>
> We demonstrate the LLM’s multi-step logical reasoning in our **Weakness 2** analysis.
>
> Due to the rebuttal length limit, we cannot include the full SAT solving traces here. More reasoning examples are provided in **Appendix K** and in the supplementary materials.
>
> ## Q2: How about SATURN vs. SFT?
>
> Our setting is RLVR, where no supervised data with intermediate CoT annotations are available; only a verifier checks whether the LLM’s final answer is correct. RL encourages the model to explore reasoning paths **on its own**, which SFT cannot achieve without costly CoT annotations, which are unavailable in our dataset.
>
> A detailed discussion is provided in **Weakness 3**.
>
> ## Q3: Would training on richer formal logic problems (e.g., QBF, theorem proving) would better?
>
> Thank you for this insightful question. We answer from two perspectives.
>
> 1. There is some work on RL training for theorem-proving tasks [4]. However, the difficulty of these problems is hard to control, which makes curriculum design challenging. This is an advantage of SAT, as we can generate large-scale instances with adjustable complexity.
>
> 2. Our work is an initial exploration of how SAT can enhance the reasoning capability of LLMs. QBF is a natural extension of SAT that incorporates quantifiers into propositional logic, and we plan to explore QBF in future work to provide richer logical problems.
>
> ## Q4: How does SAT generalize to semantic understanding or abstraction tasks?
>
> Thank you for raising this important point. We agree that semantic understanding is an important capability of the LLM. However, the RLVR community (e.g., Deepseek-R1 [5], OpenAI O-series model [6]) mainly focuses on evaluating the reasoning capability of LLMs, with benchmarks in domains such as math and code. Semantic understanding or other tasks are largely out of scope for current RLVR studies, though future work may explore it.
>
> ## Q5: Have you tested model behavior on harder SAT instances?
>
> We sincerely thank the reviewer for this clear and important question. Yes, we have evaluated SATURN-2.6k (Test Set) on SAT problems beyond 3-SAT, including 4-SAT and 5-SAT. We selected 10 categories of problems with 100 instances each, totaling 1,000 problems.
>
> **📊 Selected SAT Problem Difficulty $(n, k, l)$**
> | $n$ | 3 | 3 | 3 | 3 | 3 | 4 | 4 | 4 | 6 | 5 |
> |:---:|:----:|:---:|:---:|:---:|:---:|:----:|:--:|:---:|:---:|:---:|
> | **$k$** | **7** | **5** | **5** | **6** | **7** | **7** | **8** | **7** | **7** | **8** |
> | **$l$** | **40** | **25** | **20** | **20** | **20** | **40** | **40** | **20** | **40** | **40** |
>
> The pass@3 results are shown below:
>
> **📊  Full pass@3 results on SATURN-2.6k (Test Set)**
>
> | Model / $(n, k, l)$            | (3,7,40) | (4,7,40) | (4,8,40) | (5,8,40) | (6,7,40) | (3,5,25) | (4,7,20) | (3,7,20) | (3,6,20) | (3,5,20) | Avg.  |
> |----|------|-----|----|------|-----|------|-----|----|------|-----|-----|
> | R1-Distill-1.5B  | **1.0**  | 3.3      | 4.7      | 12.5     | 28.9     | 3.8      | 19.3     | 9.3      | 9.1      | 9.4      | 10.1  |
> | **SATURN-1.5B**  | 0.8      | **9.5**  | **11.3** | **31.7** | **62.5** | **15.8** | **50.3** | **21.6** | **18.5** | **19.5** | **24.2** |
> | R1-Distill-7B    | 6.5      | 10.4     | 11.8     | 31.3     | 56.2     | 31.2     | 65.6     | 49.9     | 52.8     | 45.6     | 36.1  |
> | **SATURN-7B**    | **22.9** | **24.3** | **26.5** | **45.7** | **72.4** | **78.2** | **91.0** | **93.3** | **93.5** | **94.1** | **64.2** |
>
> More detailed experimental results can be found in Appendix G of the paper.
>
> ## Weakness 1: Overly simple SAT datasets
> Thank you for your comment. We address the concern as follows:
>
> 1.**Difficulty should match LLM's capability.** As shown in our RQ1 experiments, even SAT instances far from the classical phase transition remain non‑trivial for LLMs. Simply increasing SAT hardness would not better build reasoning capabilities, if the LLM cannot yet solve easier cases.
>
> 2.**LLMs are not simply doing pattern matching.** Appendix L, the supplementary materials, and CoT examples provided in **Weakness 2** demonstrate that the models perform strictly step‑by‑step, clause-by-clause reasoning, rather than pattern matching.
>
> 3.**Curriculum learning ensures effective training.** Our framework strictly progresses from easy to hard, enabling both models from 1.5B to 7B parameters to gradually build reasoning capabilities.
>
> ## Weakness 2: Whether SAT is suitable for reasoning skills
>
> We appreciate the reviewer’s professional question regarding. We agree that understanding the underlying mechanisms is important. To this end, we offer key supporting evidence.
>
> A recent study, _Cognitive Behaviors that Enable Self-Improving Reasoners, or, Four Habits of Highly Effective STaRs_ [1], identifies four core habits shared by expert human reasoners and LLMs:  (1) **verification**, (2) **backtracking**, (3) **subgoal setting**, and (4) **backward chaining**. These behaviors are domain-agnostic and fundamental for solving general reasoning tasks. And [2] proposed the same concept of meta-ability. The meta-ability represent general reasoning patterns that can be generalized across SAT, math, and programming tasks. In the following, we present additional case studies to illustrate how SATURN works.
>
> **Case 1**
>
> To better understand how SATURN promotes these behaviors, we provide a case study comparing SATURN and a baseline model on the same SAT instance:
>
> > ¬A∨¬B∨¬D ∧ ¬B∨¬C∨D ∧ A∨¬B∨¬C ∧ A∨C∨E ∧ B∨C∨D
>
> R1-Distill-1.5B fails on this instance. Its trace is shown below:
>
> ```
> Wait, in the solution, D=0, so clause 3: A ∨ ¬B ∨ ¬C → 0 ∨ 0 ∨ 1 → True.
>
> Yes, that's correct.
>
> Clause 4: A ∨ C ∨ E → 0 ∨ 0 ∨ 0 → True.
>
> Clause 5: B ∨ C ∨ D → 0 ∨ 0 ∨ 0 → True.
>
> So, I think that's a valid solution.
> ```
>
> The model skips verification of earlier clauses and fails to backtrack after contradictions. The reasoning is shallow and linear. It does not apply backward chaining from goal constraints.
>
> In contrast, SATURN-1.5B produces the following trace:
>
> ```
> Hmm, that's a contradiction. So, my assumption that B=False leads to a problem with clause 1. Therefore, B cannot be false.
>
> So, B must be true. Let me backtrack.
>
> Earlier, I assumed A=True, and then B=False led to a contradiction. So, B must be true.
>
> Let me restart with A=True and B=True.
>
> Then, let's look at clause 1: ¬A ∨ ¬B ∨ ¬D. Since A and B are true, ¬A and ¬B are false, so for the clause to be true, ¬D must be true. Therefore, D must be false.
>
> So, D=False.
> ````
>
> This trace shows verification of clause satisfaction, backtracking after an invalid assumption, and backward chaining from clause structure to variable assignments. These capabilities help the model reason accurately across steps. Additional examples illustrating these reasoning behaviors are provided in Appendix K, RQ4, and the supplementary materials submitted with the paper.
>
> ## Weakness 3: Lack of RL Analysis
>
> Thank you for the valuable comment. We address your concern from two perspectives:
>
> 1. **RL improves reasoning and generalization.** Prior work (e.g., SFT Memorizes, RL Generalizes: A Comparative Study of Foundation Model Post-training) shows that SFT often overfits to training data. In contrast, RL helps the LLM self-explore and check its reasoning steps. This leads to better reasoning and generalization.
>
> 2. **RL reduces the need for data.** SFT needs carefully prepared training data with CoT anotation, which is difficult collected. However, reinforcement learning with verifiable rewards (RLVR) makes it practical to improve reasoning without extensive supervised data.
>
> ## Weakness 4: Modest improvements
>
> We clarify that our model is **not trained on additional math or code tasks**. The observed improvements therefore come from enhanced reasoning learned from SAT tasks, rather than domain transfer or data variance. This is a reliable improvement in the model’s own reasoning capability, as also illustrated in Weakness 2 and  in Appendix.
>
> Moreover, these math and code benchmarks are relatively difficult among current benchmarks. The experimental results in paper's **RQ3** show that SATURN's performance improvement exceeds prior methods.
>
> ---
>
> [1] Gandhi, Kanishk, et al. "Cognitive behaviors that enable self-improving reasoners, or, four habits of highly effective stars." arXiv preprint arXiv:2503.01307 (2025).
>
> [2] Hu, Zhiyuan, et al. "Beyond'Aha!': Toward Systematic Meta-Abilities Alignment in Large Reasoning Models." arXiv preprint arXiv:2505.10554 (2025).
>
> [3] Chu, Tianzhe, et al. "SFT Memorizes, RL Generalizes: A Comparative Study of Foundation Model Post-training." arXiv preprint arXiv:2501.17161 (2025).
>
> [4] Dong, Kefan, and Tengyu Ma. "Stp: Self-play llm theorem provers with iterative conjecturing and proving." arXiv preprint arXiv:2502.00212 (2025).
>
> [5] Guo, Daya, et al. "Deepseek-r1: Incentivizing reasoning capability in llms via reinforcement learning." arXiv preprint arXiv:2501.12948 (2025).
>
> [6] Jaech, Aaron, et al. "Openai o1 system card." arXiv preprint arXiv:2412.16720 (2024).

---

> > ### Comment · Reviewer_QZ5H · 2025-08-04
> >
> > Thank you for the detailed reply.
> >
> > While most of my concerns have been addressed, regarding W3:
> >
> > To better support the claimed benefits of RL, could you provide a direct performance comparison with an SFT-only baseline (e.g., on Math500)? Although methods specialized for a particular dataset via SFT may not fall into the same evaluation track as the one proposed in your paper, such a comparison would still offer a more convincing demonstration of the effectiveness of your method.

---

> > ### Comment · Reviewer_np9g · 2025-08-05
> >
> > Thank you for your response. I'm sure you have addressed my concerns. I will keep my score of 4.

---

> ### Author Response · Authors · 2025-08-04
>
> We sincerely appreciate your follow-up comment.
>
> Since our SAT datasets cannot provide CoT annotations due to their result verification nature, we trained an SFT-only baseline on the **Math training dataset** [1], which is the source of the Math500 benchmark and contains step by step problem solutions:
>
> ```
> {
>   "input": "How many nonzero terms are in the expansion of $$(x+4)(2x^2+3x+9)-3(x^3-2x^2+7x)?$$
> Please reason step by step, and put your final answer within \\boxed{}.",
>   "output": "First, we multiply a degree-$1$ term and a degree-$2$ term, so we have a degree-$3$ polynomial. We subtract a constant times a degree-$3$ polynomial, so we can have at most a degree-$3$ polynomial, so at most $4$ terms. However, we are unsure if any of the terms will subtract to zero, so we must multiply out the polynomials:  \\begin{align*}\n&(x+4)(2x^2+3x+9)-3(x^3-2x^2+7x)\\\\\n&\\qquad=x(2x^2+3x+9)+4(2x^2+3x+9)-(3x^3-6x^2+21x)\\\\\n&\\qquad=2x^3+3x^2+9x+8x^2+12x+36-(3x^3-6x^2+21x)\\\\\n&\\qquad=2x^3+11x^2+21x+36-(3x^3-6x^2+21x)\\\\\n&\\qquad=2x^3-3x^3+11x^2+6x^2+21x-21x+36\\\\\n&\\qquad=-x^3+17x^2+36.\n\\end{align*}As we can see, the linear term drops out, and we are left with $\\boxed{3}$ terms.",
>   "answer": "\\boxed{3}"
> }
> ```
>
> We follow the template of DeepSeek-R1-Distill-Qwen-1.5B to construct the training data:
>
> ```
> <｜begin▁of▁sentence｜><｜User｜>{input}<｜Assistant｜><think>{output}<\think>\n\n{answer}
> ```
>
> We use the TRL framework [2] for SFT. To ensure fairness in dataset size, we randomly select the most difficult Level-5 1,000 problems from the math training set for one epoch of SFT. The training hyperparameters are:
>
> | Hyperparameter                | Value  |
> |---|:---:|
> | Per‑device Batch Size         | 1      |
> | Gradient Accumulation Steps   | 4      |
> | Max Sequence Length           | 4096   |
> | Learning Rate                 | 5e‑6   |
> | Weight Decay                  | 0.01   |
> | Epoch                 | 1    |
>
> The results are shown below:
>
> | Model | AIME24/25 | AMC22/23 | Math500 | GPQA-D | LiveCodeBench | Avg |
> |-------|:--------:|:--------:|:------:|:-----:|:-------------:|:---:|
> | R1-Distill-1.5B | 21.6 | 65.1 | 83.6 | 30.3 | 16.4 | 43.4 |
> | R1-Distill-1.5B + SFT (MATH [1], 1000K) | 25.0 | 68.7 | 82.0 | 34.3 | ***14.6*** | 44.9 |
> | **Saturn-1.5B (SAT, 1000K)** | **28.3** | **73.5** | **84.6** | **37.4** | **17.4** | **48.2** |
>
> Consistent with the observations in *SFT Memorizes, RL Generalizes* [3], SFT improves performance on math related benchmarks (AIME, AMC, and Math500) that are similar to its supervised training domain. However, it slightly degrades on **LiveCodeBench**, exhibiting alignment tax [4], where SFT to a narrow domain compromises performance on other tasks. In contrast, RL training on SAT (Saturn) improves performance across all benchmarks, reflecting enhanced general reasoning.
>
> Finally, thank you again for your valuable comment. Please let us know if further details are needed.
>
> **References**
>
> [1] Hendrycks, Dan, et al. *“Measuring Mathematical Problem Solving with the MATH Dataset.”* NeurIPS, 2021.
>
> [2] von Werra, Leandro, et al. *TRL: Transformer Reinforcement Learning.* GitHub, https://github.com/huggingface/trl.
>
> [3] Chu, Tian, et al. *“SFT Memorizes, RL Generalizes: A Comparative Study of Foundation Model Post Training.”* arXiv preprint arXiv:2501.17161, 2025.
>
> [4] Ouyang, Long, et al. "Training language models to follow instructions with human feedback." Advances in neural information processing systems 35 (2022): 27730-27744.

---

> > ### Comment · Reviewer_QZ5H · 2025-08-04
> >
> > Thank you for your response.  I'm sure you have addressed my concerns.

---

> > > ### Author Response · Authors · 2025-08-05
> > >
> > > Dear Reviewer QZ5H,
> > >
> > > Thank you for your kind follow-up and for confirming that our response addressed your concerns. We sincerely appreciate the time and effort you have devoted to reviewing our work.
> > >
> > > Best regards,
> > > Authors

---

> > > ### Comment · Reviewer_QZ5H · 2025-08-05
> > >
> > > Thank you for your response. I'm sure you have addressed my concerns. Your reply clarified the theoretical motivations and provided additional empirical evidence, which helped address my earlier concerns regarding dataset simplicity, the suitability of SAT for reasoning, and the role of RL. I now find the claims more substantiated and have raised my score to 4.

---

> > > > ### Author Response · Authors · 2025-08-05
> > > >
> > > > Dear Reviewer QZ5H,
> > > >
> > > > Thank you so much for your thoughtful follow-up and for raising your score! We are truly grateful for your recognition of our work!
> > > >
> > > > Your support and constructive feedback have been tremendously helpful in improving the paper, and we sincerely appreciate your encouragement.
> > > >
> > > > Best regards, Authors

---

### Official Review · Reviewer_np9g · 2025-07-02

**Clarity:** 3
**Significance:** 3
**Originality:** 3
**Rating:** 4
**Confidence:** 3

**Summary:**

This paper introduces SATURN, a novel method for improving the reasoning abilities of large language models (LLMs) by training them on a curriculum of Boolean satisfiability (SAT) problems. The core idea is to leverage the structured, symbolic nature of SAT problems to enhance the underlying reasoning skills of LLMs. The authors propose a reinforcement learning framework where the LLM is trained on a progressively more difficult sequence of SAT problems. The difficulty of these problems is determined by a novel metric that estimates the size of the search space. A key contribution is the demonstration that training on this curriculum of SAT problems improves not only the model's ability to solve SAT problems but also its performance on a range of other reasoning tasks, including mathematical reasoning, code generation, and logical puzzles. The authors show that SATURN-trained models outperform baseline models of similar size and even some much larger models on several benchmarks. The paper also provides an in-depth analysis of the model's behavior, showing that the training process enhances the LLM's ability to perform self-verification and to generate more structured and logical reasoning step

**Questions:**

1. Could you provide more intuition or empirical validation for the chosen difficulty metric? Have you explored other possible metrics, and if so, how did they compare? A more in-depth justification for the chosen metric would help to clarify why it is particularly well-suited for this task.

2. The paper provides some excellent case studies showing how SATURN enhances the model's ability to perform self-verification. However, it would be helpful to have a more detailed analysis of the specific reasoning mechanisms that are being improved. For example, is the model learning to better understand logical connectives, or is it learning to perform a more systematic search? A more fine-grained analysis of the model's reasoning process would provide valuable insights into how SATURN works and helps.

3. The paper notes that the performance of the model plateaus after a certain point in the curriculum. Have you investigated any strategies for overcoming this plateau? For example, would it be possible to create a hybrid curriculum that integrates SAT problems with other types of reasoning tasks? This could help to further improve the model's general reasoning abilities.

**Ethical Concerns:**

["NO or VERY MINOR ethics concerns only"]

**Final Justification:**

I think the authors have addressed more of my initial concerns, but given the concerns of other reviewers, I will keep my score of 4.

**Limitations:**

Controllable Difficulty seems to make sense. However, it should be discussed with different kinds of metrics that define the difficulty.

**Paper Formatting Concerns:**

Seems no issues here.

**Quality:**

3

**Strengths And Weaknesses:**

## Strengths
- Clarity: The paper is well-written and easy to follow. The problem statement is clear, the SATURN framework is explained in detail, and the research questions are well-defined. The authors have done an excellent job of presenting their work in a way that is accessible to a broad audience, even those who may not be experts in SAT solvers or reinforcement learning.

- Originality: The paper presents a novel and significant contribution by using SAT problems as a training ground for improving the general reasoning abilities of LLMs. This is a departure from the more common approach of training on vast amounts of text data, and it opens up a new avenue for research into how to best instill abstract reasoning skills in LLMs.

- Quality: The experimental design is sound, and the authors have conducted a thorough set of experiments to validate their claims. The inclusion of extensive ablation studies in the appendix strengthens the paper by showing the impact of different components of the SATURN framework. The authors have also released their code, data, and models, which is a significant contribution to the community and allows for the reproduction and extension of their work.

##  Weaknesses

- Transferability: A key weakness is the limited transferability of the reasoning skills learned from SAT problems to more knowledge-intensive domains. While the paper shows improvements on a range of reasoning tasks, the improvements are most significant on tasks that are structurally similar to SAT problems. This suggests that the model may be learning to solve a particular type of reasoning problem, rather than developing a more general reasoning ability.

- Justification for Difficulty Metric: The paper proposes a novel metric for estimating the difficulty of SAT problems, but the justification for this particular metric is not entirely convincing. While the authors provide some intuition and empirical validation, it is not clear why this metric is superior to other possible metrics. A more thorough discussion of the design choices behind the difficulty metric would strengthen the paper.

---

> ### Author Rebuttal · Authors · 2025-07-30
>
> We are grateful for your detailed review and valuable suggestions. Each point you raised is addressed below.
>
> ## Q1: Could you provide more intuition or empirical validation for the chosen difficulty metric?
>
> Thank you for your insightful and professional question. In our early exploration, we conducted an ablation study of the SAT difficulty estimation metric on our SATURN‑2.6K test set. The dataset contains 10 categories of problems with 100 instances each, totaling 1,000 problems (Appendix G).
>
> **📊 Selected SAT Problem Difficulty $(n, k, l)$**
> | $n$ | 3 | 3 | 3 | 3 | 3 | 4 | 4 | 4 | 6 | 5 |
> |:-:|:-:|:-:|:-:|:-:|:-:|:-:|:-:|:-:|:-:|:-:|
> | **$k$** | **7** | **5** | **5** | **6** | **7** | **7** | **8** | **7** | **7** | **8** |
> | **$l$** | **40** | **25** | **20** | **20** | **20** | **40** | **40** | **20** | **40** | **40** |
>
> The pass@3 results are shown below:
>
> **📊  Full pass@3 results on SATURN-2.6k (Test Set)**
>
> | Model / $(n, k, l)$ | (3,7,40) | (4,7,40) | (4,8,40) | (5,8,40) | (6,7,40) | (3,5,25) | (4,7,20) | (3,7,20) | (3,6,20) | (3,5,20) | Avg. |
> |-|-|-|-|-|-|-|-|-|-|-|-|
> | R1-Distill-1.5B | **1.0** | 3.3 | 4.7 | 12.5 | 28.9 | 3.8 | 19.3 | 9.3 | 9.1 | 9.4 | 10.1 |
> | **SATURN-1.5B** | 0.8 | **9.5** | **11.3** | **31.7** | **62.5** | **15.8** | **50.3** | **21.6** | **18.5** | **19.5** | **24.2** |
> | R1-Distill-7B | 6.5 | 10.4 | 11.8 | 31.3 | 56.2 | 31.2 | 65.6 | 49.9 | 52.8 | 45.6 | 36.1 |
> | **SATURN-7B** | **22.9** | **24.3** | **26.5** | **45.7** | **72.4** | **78.2** | **91.0** | **93.3** | **93.5** | **94.1** | **64.2** |
>
> We then performed linear regression between each difficulty metric and model pass@3 (averaged across four models, with variance reported). The metrics capture different aspects including sparsity and structural complexity of SAT, as derived in Appendix C. The results are as follows:
>
> **📊 Metric Comparison Table**
>
> | Metric Formula | R1-Qwen-1.5B | Saturn-1.5B | R1-Qwen-7B | Saturn-7B | Avg. | Std. Dev. |
> |:--------------:|:------------:|:-----------:|:----------:|:---------:|:----:|:---------:|
> |$-k - l \cdot \log_2(1 - \frac{1}{2^n})$|0.507|0.537|0.132|0.000|0.294|0.269|
> |$\alpha \cdot \frac{k}{n} + \beta \cdot \log_2(l)$, $\alpha{=}2$, $\beta{=}1$|0.428|0.478|0.568|0.508|0.496|0.059|
> |$\alpha \cdot \frac{k}{n} + \beta \cdot \log_2(l)$, $\alpha{=}1$, $\beta{=}1$|0.240|0.279|0.719|0.826|0.516|0.300|
> |$\log_2(k \cdot l) - \log_2(2^{n} - 1)$|0.875|0.893|0.451|0.157|0.594|0.356|
> |$\log_2(k) + 2 \cdot \log_2(l) - n + \frac{k}{n}$|0.707|0.746|0.724|0.501|**0.670**|0.113|
>
> Our metric $\log_2(k) + 2 \cdot \log_2(l) - n + \frac{k}{n}$ combines both solution sparsity and structural complexity, and achieves the best overall correlation. The ratio between the solution space and the LLM’s search space is the most crucial aspect. **On SATURN‑7B, the R² value is about 0.5 because pass@3 already achieves over 90% on easy problems**, which limits the observable linear correlation in that range. Due to rebuttal length constraints, the corresponding figures are shown in Appendix Figures 9 and 10.
>
> We will include this ablation study in future revisions to strengthen the paper. Thank you again for your valuable suggestion.
>
> ## Q2: How does SATURN work?
>
> We appreciate the reviewer’s professional question regarding why training on SAT problems helps generalization to other domains such as math and code. We agree that understanding the underlying mechanisms is important. To this end, we offer key supporting evidence.
>
> A recent study, _Cognitive Behaviors that Enable Self-Improving Reasoners, or, Four Habits of Highly Effective STaRs_ [1], identifies four core habits shared by expert human reasoners and LLMs:  (1) **verification**, (2) **backtracking**, (3) **subgoal setting**, and (4) **backward chaining**. These behaviors are domain-agnostic and fundamental for solving general reasoning tasks. And [2] proposed the same concept of meta-ability. The meta-ability represent general reasoning patterns that can be generalized across SAT, math, and programming tasks. In the following, we present additional case studies to illustrate how SATURN works.
>
> **Case 1**
>
> To better understand how SATURN promotes these behaviors, we provide a case study comparing SATURN and a baseline model on the same SAT instance:
>
> > ¬A∨¬B∨¬D ∧ ¬B∨¬C∨D ∧ A∨¬B∨¬C ∧ A∨C∨E ∧ B∨C∨D
>
> R1-Distill-1.5B fails on this instance. Its trace is shown below:
>
> ```
> Wait, in the solution, D=0, so clause 3: A ∨ ¬B ∨ ¬C → 0 ∨ 0 ∨ 1 → True.
>
> Yes, that's correct.
>
> Clause 4: A ∨ C ∨ E → 0 ∨ 0 ∨ 0 → True.
>
> Clause 5: B ∨ C ∨ D → 0 ∨ 0 ∨ 0 → True.
>
> So, I think that's a valid solution.
> ```
>
> The model skips verification of earlier clauses and fails to backtrack after contradictions. The reasoning is shallow and linear. It does not apply backward chaining from goal constraints.
>
> In contrast, SATURN-1.5B produces the following trace:
>
> ```
> Hmm, that's a contradiction. So, my assumption that B=False leads to a problem with clause 1. Therefore, B cannot be false.
>
> So, B must be true. Let me backtrack.
>
> Earlier, I assumed A=True, and then B=False led to a contradiction. So, B must be true.
>
> Let me restart with A=True and B=True.
>
> Then, let's look at clause 1: ¬A ∨ ¬B ∨ ¬D. Since A and B are true, ¬A and ¬B are false, so for the clause to be true, ¬D must be true. Therefore, D must be false.
>
> So, D=False.
> ````
>
> This trace shows verification of clause satisfaction, backtracking after an invalid assumption, and backward chaining from clause structure to variable assignments. These capabilities help the model reason accurately across steps.
>
> **Case 2**
>
> We observe the same behaviors in LiveCodeBench.
>
> > You are given an array of positive integers `nums`.
> > An array is called product equivalent if the product of all elements equals `gcd(arr) * lcm(arr)`.
> > Return the length of the longest product equivalent subarray.
> > A subarray is a contiguous non-empty sequence of elements in the array.
>
> The R1-Distill-7B gives the following incorrect CoT:
>
> ```python
> Initialize current_gcd to 0 (or None)\n    Initialize current_lcm to 1\n    For each j from i to len(nums)-1:\n        current_product *= nums[j]\n        current_gcd = gcd(current_gcd, nums[j]) if current_gcd is not None else nums[j]\n        current_lcm = lcm(current_gcd, nums[j]) if current_gcd is not None else nums[j]\n
> ...
> class Solution:
>     def maxLength(self, nums: List[int]) -> int:
>         max_len = 0
>         n = len(nums)
>         for i in range(n):
>             current_product = 1
>             current_gcd = nums[i]
>             current_lcm = nums[i]
>             for j in range(i, n):
>                 current_product *= nums[j]
>                 current_gcd = math.gcd(current_gcd, nums[j])
>                 # Incorrect: uses current_gcd instead of current_lcm
>                 current_lcm = math.lcm(current_gcd, nums[j])
>                 if current_product == current_gcd * current_lcm:
>                     current_length = j - i + 1
>                     if current_length > max_len:
>                         max_len = current_length
>         return max_len
> ````
>
> This model incorrectly updates `current_lcm` using `current_gcd`. It fails to verify whether the condition is preserved after each step. It also does not backtrack or revise when the invariant breaks.
>
> SATURN-7B corrects the error with a single key change:
>
> ```python
> current_lcm = (current_lcm * nums[j]) // math.gcd(current_lcm, nums[j])
> ```
>
> This update uses the previous LCM, maintaining the correct invariant. The change reflects step-by-step verification and inference from the problem condition. The model traces through the logic and only updates when valid.
>
> In summary, SATURN shows consistent use of verification, backtracking, and backward chaining across general reasoning tasks. These reasoning strategies, learned from SAT, enable generalization to unfamiliar domains through principled logical behavior. Additional examples illustrating these reasoning behaviors are provided in Appendix K and the supplementary materials submitted with the paper.
>
> ## Q3: Have you investigated any strategies for overcoming the plateau?
>
> Your question is very professional and forward-looking. Thank you for raising it.
>
> Following the insights from the DeepScaleR [3], we explored extending the RL context length from 8k to 12k and then to 16k tokens to overcome the plateau. The effect was limited, and extremely long responses can trigger OOM under our compute constraints.
>
> Our experiments confirm that SAT problems serve as an effective complement to math and code tasks. The plateau may be addressed through mixed hybrid curriculum training with SAT and math data. Due to time constraints, we conducted a very simple experiment: We fine-tuned SATURN-1.5B using RL for 50 steps on the GSM8K dataset. The results are as follows:
>
> | Model                  | AIME 24 | AIME 25 | Math500 | GPQA-D |
> |--|---:|----:|----:|---:|
> | SATURN-1.5B            |   30.0  |   26.7  |   84.6  |   37.4 |
> | SATURN-1.5B + GSM8K    |   **36.7**  |   **30.0**  |   **85.8**  |   **37.9** |
>
> The idea of a hybrid curriculum with other reasoning tasks is excellent, and we will focus on it carefully in our future work.
>
> ## Weakness 1: Transferability
>
> Please see the response to **Q2**.
>
> ## Weakness 2: Justification for Difficulty Metric
>
> Please see the response to **Q1**.
>
> Finally, we thank you for your thoughtful review and your recognition of our efforts.
>
> ------
>
> [1] Gandhi, Kanishk, et al. "Cognitive behaviors that enable self-improving reasoners, or, four habits of highly effective stars." arXiv preprint arXiv:2503.01307 (2025).
>
> [2] Hu, Zhiyuan, et al. "Beyond'Aha!': Toward Systematic Meta-Abilities Alignment in Large Reasoning Models." arXiv preprint arXiv:2505.10554 (2025).
>
> [3] Luo, Michael, et al. DeepScaleR: Surpassing O1-Preview with a 1.5B Model by Scaling RL. 2025. Notion Blog.

---

> ### Author Response · Authors · 2025-08-05
>
> Dear **Reviewer np9g**,
>
> Thank you so much for your follow-up comment. We truly appreciate your recognition of our work and your dedicated review, which have been very helpful for improving our paper.
>
> Please feel free to share any further thoughts or questions.
>
> ---
>
> >  *Thank you for your response. I'm sure you have addressed my concerns. I will keep my score of 4.*
>
> As a small note, we noticed that your comment appears under another reviewer’s thread, just in case it was unintentional.
>
> Best regards, Authors

---

### Official Review · Reviewer_SUsx · 2025-07-03

**Clarity:** 3
**Significance:** 3
**Originality:** 3
**Rating:** 5
**Confidence:** 3

**Summary:**

The authors propose SATURN, a RL framework intended to improve the reasoning capabilities of LLMs with SAT problems. Towards this end, they construct SAT problems (SATURN 2.6k dataset) of increasing difficulty to create a curriculum of reasoning problems of increasing difficulty. This is then interleaved with a LLM RL training loop (using GRPO) to improve LLM reasoning ability. They apply the framework to DeepSeek-R1-Distill-Qwen-1.5B and 7B, and demonstrate that the application of SATURN can improve the performance of the base LLMs on SAT problems. They also demonstrate that it can generalize to Math and Programming questions as well as complement other approaches to improve reasoning in LLMs.

**Questions:**

1. Do the authors have an explanation as to why such a method, even though it is trained on SAT, should generalize to other settings? For instance, does it encourage the LLMs to consider more cases and/or negations that help it reason more effectively in math/coding domains?
2. Have you tested SAT problems that are not given as 3-SAT problems? I wonder about data leakage as the models have likely picked up instances of 3-SAT problems during pre-training. However, I would imagine that moderate-size SAT problems would be less prevalent on the internet since they are too large to be instructive or solvable without being converted into 3-SAT.
3. Why does Saturn-7B perform worse on AIME24/25 compared to DeepSeek-R1-Distill-Qwen-7B?
4. What does the lengthening phenomenon look like for a curriculum approach vs. a mixed training approach? While I appreciate the difficulty of performing all these ablations, I am curious, since curriculum learning adds another layer of complexity to the post-training step. Does mixed-training result in shorter responses compared to a curriculum approach?

**Ethical Concerns:**

["NO or VERY MINOR ethics concerns only"]

**Final Justification:**

The discussion with the reviewers has helped answer some concerns on why SATURN. The two case studies that highlight LLM reasoning: (1) verification, (2) backtracking, (3) subgoal setting, and (4) backward chaining, should be put in appendix somewhere in the paper to help the reader understand why SATURN can help the reasoning process, and hopefully, inform future systematic study. Additional results, such as how the response length differs between curriculum learning and mixed training, would also be interesting to be put in the paper.

Overall, I found the method to be promising (at the scale it is tested on). Nevertheless, provided that the code is open source and helps other researchers build on improving and analyzing LLM reasoning, I am leaning towards accepting this paper and am maintain my rating.

**Limitations:**

The authors correctly highlight limitations in Section 4.1.

**Quality:**

3

**Strengths And Weaknesses:**

The paper works on an important problem—new methods to improve reasoning capabilities in LLMs. The proposed approach of combining curriculum learning and RL is natural. The study also is ablated (e.g., testing the efficacy of mixed training vs. curriculum) and comprehensive (e.g., testing transferability of SAT training to math/programming). Lastly, I found it promising that training on SAT problems could improve reasoning in other domains.

While I believe the methods and study are sound, I found myself wishing for more explanation as to why such a method should work and for a deeper analysis of the reasoning chains produced as a result of training with SATURN. For instance, is it because training on SAT problems causes the model to take smaller reasoning steps, enumerate more cases, or something else, that also enables it to generalize better to math/coding problems?

---

> ### Author Rebuttal · Authors · 2025-07-30
>
> Thank you very much for your helpful comments. We respond to your concerns point by point in the following.
>
> ## Q1: Why does SATURN work?
>
> We appreciate the reviewer’s professional question regarding why training on SAT problems helps generalization to other domains such as math and code. We agree that understanding the underlying mechanisms is important. To this end, we offer key supporting evidence.
>
> A recent study, _Cognitive Behaviors that Enable Self-Improving Reasoners, or, Four Habits of Highly Effective STaRs_ [1], identifies four core habits shared by expert human reasoners and LLMs:  (1) **verification**, (2) **backtracking**, (3) **subgoal setting**, and (4) **backward chaining**. These behaviors are domain-agnostic and fundamental for solving general reasoning tasks. And [2] proposed the same concept of meta-ability. The meta-ability represent general reasoning patterns that can be generalized across SAT, math, and programming tasks. In the following, we present additional case studies to illustrate how SATURN works.
>
> **Case 1**
>
> To better understand how SATURN promotes these behaviors, we provide a case study comparing SATURN and a baseline model on the same SAT instance:
>
> > ¬A∨¬B∨¬D ∧ ¬B∨¬C∨D ∧ A∨¬B∨¬C ∧ A∨C∨E ∧ B∨C∨D
>
> R1-Distill-1.5B fails on this instance. Its trace is shown below:
>
> ```
> Wait, in the solution, D=0, so clause 3: A ∨ ¬B ∨ ¬C → 0 ∨ 0 ∨ 1 → True.
>
> Yes, that's correct.
>
> Clause 4: A ∨ C ∨ E → 0 ∨ 0 ∨ 0 → True.
>
> Clause 5: B ∨ C ∨ D → 0 ∨ 0 ∨ 0 → True.
>
> So, I think that's a valid solution.
> ```
>
> The model skips verification of earlier clauses and fails to backtrack after contradictions. The reasoning is shallow and linear. It does not apply backward chaining from goal constraints.
>
> In contrast, SATURN-1.5B produces the following trace:
>
> ```
> Hmm, that's a contradiction. So, my assumption that B=False leads to a problem with clause 1. Therefore, B cannot be false.
>
> So, B must be true. Let me backtrack.
>
> Earlier, I assumed A=True, and then B=False led to a contradiction. So, B must be true.
>
> Let me restart with A=True and B=True.
>
> Then, let's look at clause 1: ¬A ∨ ¬B ∨ ¬D. Since A and B are true, ¬A and ¬B are false, so for the clause to be true, ¬D must be true. Therefore, D must be false.
>
> So, D=False.
> ````
>
> This trace shows verification of clause satisfaction, backtracking after an invalid assumption, and backward chaining from clause structure to variable assignments. These capabilities help the model reason accurately across steps.
>
> **Case 2**
>
> We observe the same behaviors in LiveCodeBench.
>
> > You are given an array of positive integers `nums`.
> > An array is called product equivalent if the product of all elements equals `gcd(arr) * lcm(arr)`.
> > Return the length of the longest product equivalent subarray.
> > A subarray is a contiguous non-empty sequence of elements in the array.
>
> The R1-Distill-7B gives the following incorrect CoT:
>
> ```python
> Initialize current_gcd to 0 (or None)\n    Initialize current_lcm to 1\n    For each j from i to len(nums)-1:\n        current_product *= nums[j]\n        current_gcd = gcd(current_gcd, nums[j]) if current_gcd is not None else nums[j]\n        current_lcm = lcm(current_gcd, nums[j]) if current_gcd is not None else nums[j]\n
> ...
> class Solution:
>     def maxLength(self, nums: List[int]) -> int:
>         max_len = 0
>         n = len(nums)
>         for i in range(n):
>             current_product = 1
>             current_gcd = nums[i]
>             current_lcm = nums[i]
>             for j in range(i, n):
>                 current_product *= nums[j]
>                 current_gcd = math.gcd(current_gcd, nums[j])
>                 # Incorrect: uses current_gcd instead of current_lcm
>                 current_lcm = math.lcm(current_gcd, nums[j])
>                 if current_product == current_gcd * current_lcm:
>                     current_length = j - i + 1
>                     if current_length > max_len:
>                         max_len = current_length
>         return max_len
> ````
>
> This model incorrectly updates `current_lcm` using `current_gcd`. It fails to verify whether the condition is preserved after each step. It also does not backtrack or revise when the invariant breaks.
>
> SATURN-7B corrects the error with a single key change:
>
> ```python
> current_lcm = (current_lcm * nums[j]) // math.gcd(current_lcm, nums[j])
> ```
>
> This update uses the previous LCM, maintaining the correct invariant. The change reflects step-by-step verification and inference from the problem condition. The model traces through the logic and only updates when valid.
>
> In summary, SATURN shows consistent use of verification, backtracking, and backward chaining across general reasoning tasks. These reasoning strategies, learned from SAT, enable generalization to unfamiliar domains through principled logical behavior. Additional examples illustrating these reasoning behaviors are provided in Appendix K and the supplementary materials submitted with the paper.
>
> ## Q2: Have you evaluated SATURN on SAT problems beyond 3‑SAT?
>
> We sincerely thank the reviewer for this clear and important question. Yes, we have evaluated SATURN-2.6k (Test Set) on SAT problems beyond 3-SAT, including 4-SAT and 5-SAT. We selected 10 categories of problems with 100 instances each, totaling 1,000 problems.
>
> **📊 Selected SAT Problem Difficulty $(n, k, l)$**
> | $n$ | 3 | 3 | 3 | 3 | 3 | 4 | 4 | 4 | 6 | 5 |
> |:-:|:-:|:-:|:-:|:-:|:-:|:-:|:-:|:-:|:-:|:-:|
> | **$k$** | **7** | **5** | **5** | **6** | **7** | **7** | **8** | **7** | **7** | **8** |
> | **$l$** | **40** | **25** | **20** | **20** | **20** | **40** | **40** | **20** | **40** | **40** |
>
> The pass@3 results are shown below:
>
> **📊  Full pass@3 results on SATURN-2.6k (Test Set)**
>
> | Model / $(n, k, l)$ | (3,7,40) | (4,7,40) | (4,8,40) | (5,8,40) | (6,7,40) | (3,5,25) | (4,7,20) | (3,7,20) | (3,6,20) | (3,5,20) | Avg. |
> |-|-|-|-|-|-|-|-|-|-|-|-|
> | R1-Distill-1.5B | **1.0** | 3.3 | 4.7 | 12.5 | 28.9 | 3.8 | 19.3 | 9.3 | 9.1 | 9.4 | 10.1 |
> | **SATURN-1.5B** | 0.8 | **9.5** | **11.3** | **31.7** | **62.5** | **15.8** | **50.3** | **21.6** | **18.5** | **19.5** | **24.2** |
> | R1-Distill-7B | 6.5 | 10.4 | 11.8 | 31.3 | 56.2 | 31.2 | 65.6 | 49.9 | 52.8 | 45.6 | 36.1 |
> | **SATURN-7B** | **22.9** | **24.3** | **26.5** | **45.7** | **72.4** | **78.2** | **91.0** | **93.3** | **93.5** | **94.1** | **64.2** |
>
> More detailed experimental results can be found in Appendix G of the paper.
>
> ## Q3: Why does Saturn-7B perform worse on AIME24/25?
>
> Thank you for the important question. We appreciate the reviewer for pointing out the performance gap on AIME24/25. We believe this drop is primarily due to the limitations discussed in **Section 5**, especially **Limitation ❶: Knowledge limitations ❸ Plasticity and Forgetting**.
>
> For instance, SATURN-7B failed on the following AIME 2024 problem:
>
> > *Let \( ABCDEF \) be a convex equilateral hexagon in which all pairs of opposite sides are parallel. The triangle whose sides are extensions of segments \( AB, CD, \) and \( EF \) has side lengths 200, 240, and 300. Find the side length of the hexagon.*
>
> After manual checking, SATURN-7B generates a mostly correct reasoning chain, but makes a small mistake at the final step by incorrectly applying the harmonic mean formula: $$s = \frac{3}{\frac{1}{200} + \frac{1}{240} + \frac{1}{300}} = 240$$ The correct relation is based on geometric decomposition: $$s \left( \frac{1}{200} + \frac{1}{240} + \frac{1}{300} \right) = 1 \quad \Rightarrow \quad s = 80$$
>
> This error shows that the model selects a plausible-looking formula. SAT enhances the reasoning ability. But due to the potential issue of catastrophic forgetting in RL, certain math knowledge may be weakened.  When the forgotten knowledge exceeds the improvement from enhanced reasoning, it appears as a drop in benchmark performance, especially on AIME, which is the hardest among these benchmarks.
>
> However, it can be addressed through mixed hybrid curriculum training with SAT and math data. Due to time constraints, we conducted a very simple experiment: We fine-tuned SATURN-1.5B using RL for 50 steps on the GSM8K dataset. The results are as follows:
>
> | Model  | AIME 24 | AIME 25 | Math500 | GPQA-D |
> |-|---:|--:|---:|--:|
> | SATURN-1.5B  |   30.0  |   26.7  |   84.6  |   37.4 |
> | SATURN-1.5B + GSM8K    |   **36.7**  |   **30.0**  |   **85.8**  |   **37.9** |
>
> In future, we will focus on hybrid curriculum with other types of reasoning tasks.
>
> ## Q4: How does response length differ between curriculum learning and mixed training?
>
> Thank you for your question. Your question is very insightful and highly professional. We are glad to address the difference between the curriculum approach and the mixed training approach. We retained the logs from the mixed training ablation on Qwen2.5-7B-Instruct-1M. Since external links and images are not allowed, we present the response length at each step as a list:
> ```
> [903.2, 1012.1, 1176.8, 1135.0, 1026.6, 842.7, 836.2, 852.7, 846.8, 852.6, 823.7, 777.5, 754.6, 769.4, 875.6, 943.7, 865.6, 803.7, 839.3, 793.5, 869.5, 926.2, 947.8, 961.4, 953.0]
> ```
> Since the list is too long, for clarity we report the average of every 10 consecutive numbers. As shown, during the early stage of training, the response length increases slowly. After around 50 steps, it drops sharply and struggles to grow again. As a result, the final responses of the mixed training approach are noticeably shorter compared to the curriculum approach.
>
> ## Weakness1: Why does SATURN work?
>
> Please see the response to **Q1**.
>
> Finally, your feedback is highly valuable to us, and we appreciate your recognition of our contribution.
>
> ------
>
> [1] Gandhi, Kanishk, et al. "Cognitive behaviors that enable self-improving reasoners, or, four habits of highly effective stars." arXiv preprint arXiv:2503.01307 (2025).
>
> [2] Hu, Zhiyuan, et al. "Beyond'Aha!': Toward Systematic Meta-Abilities Alignment in Large Reasoning Models." arXiv preprint arXiv:2505.10554 (2025).

---

> > ### Comment · Area_Chair_QWGt · 2025-08-05
> > **Please post your response**
> >
> > Dear Reviewer SUsx,
> >
> > This is a gentle reminder to post your response. The deadline for the author-reviewer discussion period is approaching. The authors have responded to your reviews and also to others' reviews. Please have an open discussion with the authors about your reviews and whether your concerns have been addressed.
> >
> > Best,
> >
> > AC

---

> > > ### Comment · Reviewer_SUsx · 2025-08-06
> > >
> > > Thank you for your response.
> > >
> > > Adding each of the responses to the questions would be helpful (in the appendix) to strengthen the paper.

---

> ### Author Response · Authors · 2025-08-06
>
> Dear Reviewer SUsx,
>
> Thank you for your comment and for carefully reviewing our paper. We appreciate your recognition of our work. And we will include the responses to each question in future versions.
>
> We are happy to discuss any further questions at any time.
>
> Best regards,
> Authors

---

### Note · Authors · 2025-08-12

We thank all reviewers for their valuable feedback and recognition of our contributions. Below we summarize the main strengths and concerns.
## **Strengths**
1. **Important research question**: R1 notes that our work addresses an important problem of improving reasoning capabilities in LLMs. R2 highlights that the research questions are well defined, with a presentation accessible to a broad audience.
2. **Novel training framework**: R2 emphasizes that using SAT problems as a training ground for improving general reasoning abilities is a novel and significant contribution that opens new research directions. R4 recognizes that SATURN's ability to generalize beyond SAT is interesting and significant.
3. **Solid experimental evaluation**: R2 considers the experimental design sound, with extensive ablation studies and the release of code, data, and models to support reproducibility.

## **Concerns**

Reviewers hope we: 1) provide **more detailed explanations on SATURN's generalization ability to math and programming**; 2) clarify **more detailed aspects of the method design**; and 3) express **curiosity about its performance on broader benchmarks**.

---

In our rebuttal, we address these points:

1. For **generalization**, we provide more case studies showing that SATURN fosters meta-reasoning habits such as verification, backtracking, subgoal setting, and backward chaining, which transfer across domains. We also add comparison experiments with SFT to further validate generalization.
2. For **method design**, we add ablation experiments comparing different SAT difficulty estimation metrics. The results demonstrate that our estimation metric, which combines solution sparsity and structural complexity, achieves the strongest correlation with LLMs' performance.
3. For **broader benchmarks**, we add three additional math benchmarks. We also clarify the results on more difficult SAT problems in Appendix G. These results confirm SATURN's effectiveness. Additionally, we conduct RL combining SAT with math and programming tasks, which further improves performance.

In future revisions, we will:

1. Provide finer-grained analysis of reasoning patterns.
2. Expand the explanation and ablation validation of the difficulty metric.
3. Explore hybrid curricula combining SAT with math and programming problems.

We believe SATURN is a valuable step toward building LLMs with general reasoning skills. We expect our work to contribute to research on LLM reasoning.

---

### Decision · Program_Chairs · 2025-09-17

**Decision:**

Accept (spotlight)

**Comment:**

The paper proposes SATURN, a SAT-based reinforcement learning framework to enhance LLMs' reasoning capabilities. It addresses limitations of existing RL tasks by enabling scalable SAT task construction, rule-based verification, and precise difficulty control via a curriculum learning pipeline. The authors introduce Saturn-2.6k dataset and apply SATURN to DeepSeek-R1-Distill-Qwen, achieving significant improvements on SAT problems, modest gains on math/programming tasks, and outperforming SOTA RL task construction approaches. The authors argue that SAT training fosters self-verification and structured reasoning, enabling generalization to other domains.

Reviewers acknowledge the paper’s originality in leveraging SAT problems for LLM reasoning, calling it a novel and significant contribution. The curriculum design and difficulty metric are praised as well-motivated and empirically effective, while the comprehensive evaluation—including ablation studies and transferability tests—adds robustness. The code is open source and helps other researchers build on improving and analyzing LLM reasoning. Reviewers also find the generalization to math/coding tasks interesting and significant, suggesting SATURN may improve broader reasoning skills.

On the other hand, reviewers raise concerns: (1) The SAT problems may be too simple, relying on "shallow heuristics" rather than deep reasoning; (2) Limited justification for the difficulty metric’s superiority over alternatives; (3) Under-analysis of RL’s role versus supervised fine-tuning. Some reviewers also note the "modest" downstream improvements and speculate about data leakage. The paper’s claims about general reasoning mechanisms lack empirical grounding.

In general, the paper’s strengths outweigh its weaknesses. The innovative use of SAT problems, robust curriculum design, and demonstrated improvements on multiple benchmarks justify acceptance. While critiques about metric justification and task simplicity are valid, they do not undermine the core contribution. Addressing these in revision by clarifying the difficulty metric’s rationale and adding RL-vs.-supervised comparisons would further strengthen the work. Given its potential to inspire future research, I recommend acceptance.

During the rebuttal, the authors have provided detailed explanations on reviewers' concerns, regarding dataset simplicity, the suitability of SAT for reasoning, and the role of RL. They have also provided additional empirical validation for the chosen difficulty metric and model behavior on harder SAT instances. The effective rebuttal leads to the increase of average rating.